# Mitigating Staleness in Asynchronous Pipeline Parallelism via Basis Rotation

Hyunji Jung [* 1]  Sungbin Shin [* 1]  Namhoon Lee [1]

## Abstract

Asynchronous pipeline parallelism maximizes hardware utilization by eliminating the pipeline bubbles inherent in synchronous execution, offering a path toward efficient large-scale distributed training. However, this efficiency gain can be compromised by gradient staleness, where the immediate model updates with delayed gradients introduce noise into the optimization process. Crucially, we identify a critical, yet often overlooked, pathology: this delay scales linearly with pipeline depth, fundamentally undermining the very scalability that the method originally intends to provide. We trace this pathology to a specific property of the optimization landscape: the misalignment between the Hessian eigenbasis and the standard coordinate basis, which triggers oscillations in the update trajectories of coordinate-wise adaptive optimizers. We identify that these oscillations cause delayed updates to diverge from their true counterparts, invalidating their use for current iterations. This insight is formalized through theoretical analysis, including a convergence bound showing that basis misalignment amplifies the delay penalty, and substantiated with empirical evaluation. To address this, we propose basis rotation, a framework that rotates the optimizer's coordinate system to align with the Hessian eigenbasis, keeping delayed updates useful. We theoretically demonstrate that basis rotation minimizes basis misalignment, thereby counteracting the conditions that amplify delay penalties. Empirically, in training up to a 3B-parameter LLM, basis rotation reduces the required iterations by 81.7% compared to the best-performing asynchronous baseline.[1]

[*]Equal contribution [1]POSTECH. Correspondence to: Namhoon Lee <namhoon.lee@postech.ac.kr>.

*Proceedings of the 43rd International Conference on Machine Learning*, Seoul, South Korea. PMLR 306, 2026. Copyright 2026 by the author(s).

[1]Our code is available at https://github.com/LOG-postech/basis-rotation.

## 1. Introduction

Training large-scale LLMs requires partitioning the model across multiple devices, as the memory footprint of such models far exceeds the capacity of individual accelerators. Pipeline parallelism addresses this by dividing the model into sequential stages, each allocated to a separate device. As a result, it has become a cornerstone of LLM training—alongside data, tensor, and context parallelism—as models continue to grow (Dubey et al., 2024; Adler et al., 2024; Liu et al., 2024; Yang et al., 2025; Team et al., 2025).

However, the efficiency of this approach is fundamentally constrained by its synchronous design, which mandates that each stage wait for the completion of backward passes of every other stage before updating its weights (Huang et al., 2019; Fan et al., 2021; Li & Hoefler, 2021). This dependency results in suboptimal hardware utilization by introducing significant idle periods, commonly referred to as pipeline bubbles.

Asynchronous pipeline parallelism aims to mitigate these idle periods by allowing each stage to proceed with subsequent computations without waiting for the completion of backward passes of other stages (Narayanan et al., 2019; 2021). While this approach significantly increases hardware utilization, it introduces gradient staleness as a consequence of the temporal gap between gradient calculation and application; *i.e.*, gradients arrive at the update step after the model has undergone multiple intervening weight updates (see Figure 1). This delayed arrival has emerged as a primary challenge in asynchronous training, as it often degrades convergence stability and final model performance (Yang et al., 2021; Ajanthan et al., 2025).

Notably, we find that gradient staleness presents a critical bottleneck, particularly for large models. This is because gradient delay increases alongside model size and pipeline depth; indeed, the number of stages can easily reach tens or hundreds in large-scale configurations[2]. In fact, we observe that increasing the number of stages for a fixed model results in a drastic 5.81-fold slowdown in convergence speed (see Figure 2a). This suggests that staleness is not merely a nuisance, but a fundamental barrier to the scalability of asynchronous pipelining, despite the fact that asynchronous

[2]We provide an analysis of model-stage scaling in Appendix A.

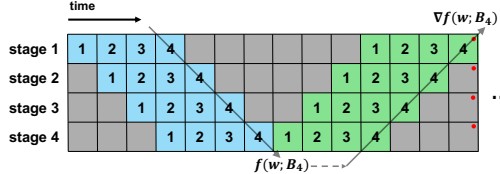

(a) Synchronous pipeline parallelism

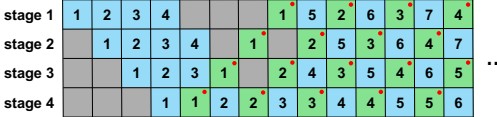

(b) Asynchronous pipeline parallelism

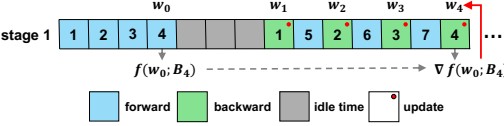

(c) Delay in asynchronous pipeline parallelism

*Figure 1.* (a–b) Schematic diagrams showing how micro-batches (blocks 1–7) are processed over time: a micro-batch travels from stage 1 to stage 4 in a forward pass (blue), and then goes back to stage 1 through a backward pass (green). Once the gradient becomes available after the backward pass, the model is updated (red dots indicate the time points). Asynchronous pipelining removes idle periods by processing subsequent micro-batches immediately after completing a backward pass without waiting for the completion of the pipeline cycle. (c) An illustration of model update with delayed gradient at stage 1: $w_3$ is updated to $w_4$ with $\nabla f(w_0; B_4)$.

pipeline parallelism aims to facilitate large-scale training.

Our analysis reveals that this degradation is deeply rooted in the interaction between delayed updates and the characteristics of Adam (Kingma & Ba, 2015), the de facto optimizer for LLM pre-training (Touvron et al., 2023b; Dubey et al., 2024; Liu et al., 2024). Specifically, we identify basis misalignment—a condition where the Hessian eigenbasis is not aligned with the standard coordinate basis—as the central reason staleness damages convergence. Since Adam's coordinate-wise adaptivity becomes ineffective under basis misalignment, the update direction changes rapidly between when a gradient is computed and when it is applied, so the delayed update no longer matches the non-delayed counterpart. We substantiate this intuition through empirical observations and convergence analysis, showing that misalignment directly amplifies the impact of staleness on convergence.

To address this, we propose mitigating the impact of staleness through basis rotation, a framework that transforms the optimization space to align the Hessian eigenbasis with the standard basis. By the transformation, it effectively leverages the curvature-aware adaptivity and straightens the

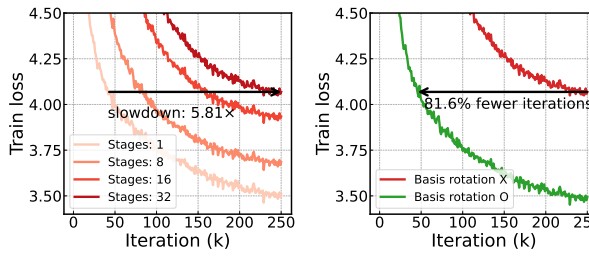

(a) Impact of delay      (b) Effectiveness of solution

*Figure 2.* Summary of this work. (a) Impact of pipeline depth (*i.e.*, number of stages) on convergence of asynchronous pipeline parallel LLM pre-training. In all cases, the model itself is kept the same while the number of stages is divided to be different. Increased delay leads to significant degradation on convergence speed. (b) Basis rotation substantially accelerates convergence in the presence of a large delay (here, for the case of 32 stages).

local trend of the update trajectory. This linearized trajectory keeps the delayed update directions better aligned with the actual optimization path. Consequently, basis rotation allows the use of delayed gradient information while mitigating staleness in the update direction. We further introduce several practical rotation strategies using different Hessian approximations and provide a theoretical analysis of their respective approximation qualities.

Empirical evaluations on LLM pre-training demonstrate that our proposed solution significantly neutralizes the degradation caused by gradient delay; for example, basis rotation helps to achieve the same training loss in up to 81.6% fewer iterations than the standard asynchronous pipeline parallel training (Figure 2b). Furthermore, introducing a stage-aware rotation strategy that allocates computational budget proportionally to per-stage delay yields an additional 29.2% speedup. These results suggest that basis rotation is a vital component for enabling high-fidelity, large-scale asynchronous pipeline parallelism.

Our key contributions are summarized as follows:
- Identifying a critical issue in scaled pipelines: We show that asynchronous pipeline parallel training suffers from significant convergence and model performance degradation as the number of stages increases, which has received limited formal study in the literature.
- Demystifying the mechanism of failure: We point out basis misalignment as the primary reason Adam-type optimizers are sensitive to delay. We provide both empirical evidence and theoretical convergence analysis to show how this misalignment exacerbates the negative impact of stale gradients (Section 2).
- The basis rotation framework: To mitigate these effects, we propose basis rotation, a method designed to realign the optimization trajectory and counteract the delay-induced instability under basis misalignment (Section 3).
- Empirical validation: We provide extensive experimental

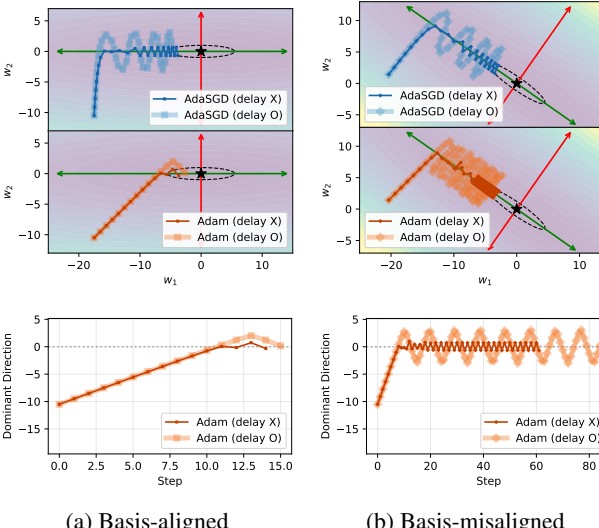

*Figure 3.* Impact of basis alignment on the effect of delay. (Top) Trajectories of AdaSGD and Adam with and without delay. (Bottom) Update of Adam along the dominant direction (red arrows in the top panel). (a) When the Hessian eigenbasis is aligned with the standard coordinate basis, Adam yields a stable trajectory and robust to delay; under misalignment (b), it oscillates along the dominant direction, and training suffers from delay. See Appendix D.1 for the explanation of the use of AdaSGD and experimental details.

results, including 3B-scale pre-training, demonstrating that our solution effectively alleviates the performance penalties inherent in asynchronous pipelines (Section 4).

## 2. Understanding the Impact of Delay

While asynchronous pipeline parallelism offers significant throughput advantages, we find that the resulting gradient delay can lead to pathological training behaviors where increasing model scale can critically damage model performance. To resolve this issue, we begin by understanding the fundamental mechanisms of how delayed gradients interact with the optimization landscape. We attribute the degradation from delay primarily to basis misalignment. Under such conditions, coordinate-wise adaptive optimizers such as Adam become vulnerable to delays.

### 2.1. Root of Degradation: Basis Misalignment

The heterogeneous curvature of Transformer loss landscapes renders a single global step size ineffective (Zhang et al., 2024; 2025b), establishing coordinate-wise adaptive optimizers like Adam as the de facto standard (Touvron et al., 2023b; Dubey et al., 2024). By adapting step sizes for each parameter individually, these optimizers effectively mitigate the oscillations that typically arise when the step size is excessive relative to local curvature (Kingma & Ba, 2015; Zhang et al., 2020; Pan & Li, 2022).

To illustrate this effect, consider the quadratic objective

$\min_w \frac{1}{2} w^\top H w$ (see Figure 3), first without delay. When the Hessian $H$ is diagonal, as depicted in Figure 3a, AdaSGD (Wang & Wiens, 2020)—which applies a single learning rate uniformly across all coordinates—exhibits oscillatory updates along the dominant eigenvector direction (red arrows) while progressing slowly along non-dominant directions (green arrows). In contrast, Adam effectively suppresses these oscillations, yielding a nearly direct trajectory toward the optimum. However, this coordinate-wise adaptivity critically depends on the alignment between the Hessian eigenbasis and the coordinate system underlying the adaptive method. As shown in Figure 3b, under basis misalignment, Adam's effective adaptivity diminishes and its trajectory resembles the behavior of AdaSGD with severe oscillations occur along the dominant eigenvector direction.

Crucially, we identify this oscillation as the primary mechanism that amplifies sensitivity to delay. When the trajectory oscillates rapidly, delayed gradients are likely to be stale—pointing in outdated or even adversarial directions relative to the current iterate—thereby substantially degrading convergence, as in Figure 3b. On the other hand, a smooth trajectory with reduced oscillations ensures that the delayed update remains closely aligned with its non-delayed counterpart, rendering Adam robust to gradient staleness, as in Figure 3a. When the update directions $\{u_t\}$ are locally consistent (i.e., $u_t \approx u$), the delayed trajectory $\{\tilde{x}_t\}$ closely approximates the original trajectory $\{x_t\}$, provided the gradient noise is small. We briefly illustrate this via induction (see Appendix B for details). Assume $\tilde{x}_i \approx x_i$ for $i \leq k-1$. Then, the moments $\tilde{m}_k$ and $\tilde{v}_k$ computed with delayed gradients $\tilde{g}_k$ at $\tilde{x}_{k-1-\tau}$ are approximated as follows:

$$\tilde{m}_k = \beta_1 \tilde{m}_{k-1} + (1-\beta_1)\tilde{g}_k$$
$$\approx \beta_1 m_{k-\tau-1} + (1-\beta_1)g_{k-\tau} = m_{k-\tau} \quad (1)$$
$$\tilde{v}_k = \beta_2 \tilde{v}_{k-1} + (1-\beta_2)\tilde{g}_k^2$$
$$\approx \beta_2 v_{k-\tau-1} + (1-\beta_2)g_{k-\tau}^2 = v_{k-\tau}, \quad (2)$$

Consequently, the resulting update direction satisfies $\tilde{u}_k \approx u_{k-\tau} \approx u_k$, ensuring that the subsequent iterate $\tilde{x}_k$ remains aligned with its non-delayed counterpart $x_k$.

### 2.2. Empirical Observation

Building on the high-level intuition from quadratic optimization, we design a spiral loss landscape to better simulate the non-stationary geometry of deep neural networks where the Hessian eigenbasis evolves along the training trajectory. Figure 4a shows the optimization trajectory of Adam trained without delay. At each randomly selected point along this trajectory, we measure the slowdown ratio $T_{\text{delay}}/T_{\text{no-delay}}$, defined as the iteration ratio required to advance a fixed angular interval with and without delay. Each of these measurements corresponds to a specific point plotted in Figure 4b. See Appendix D.1 for further details.

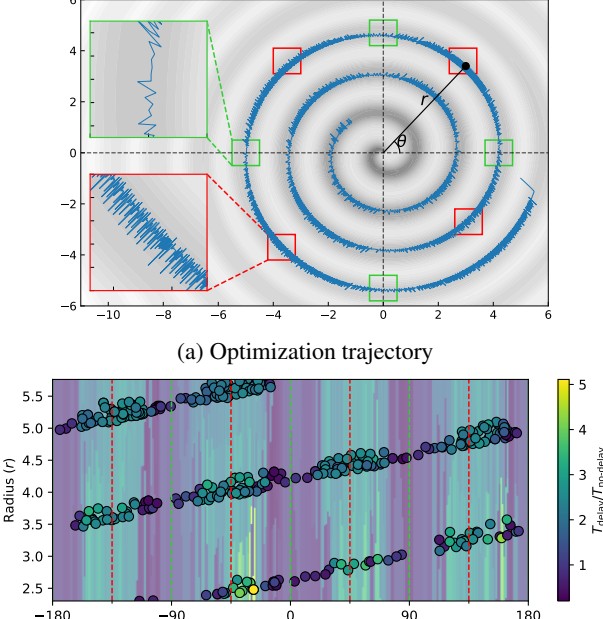

(a) Optimization trajectory

(b) Impact of delay on convergence speed

*Figure 4.* (a) Optimization trajectory of Adam on a spiral loss. The optimizer maintains a stable trajectory in basis-aligned regions (green boxes) but exhibits severe oscillations in misaligned regions (red boxes). (b) Slowdown ratio $T_{\text{delay}}/T_{\text{no-delay}}$ in different regions. The ratio is minimized near basis-aligned regions (green dotted lines), whereas it is maximized in misaligned regions (red dotted lines). This demonstrates that basis misalignment significantly amplifies the deleterious effects of delayed gradients.

As shown in Figure 4a, Adam exhibits a stable trajectory with minimal oscillation when the Hessian eigenbasis aligns with the standard coordinate basis (see green boxes). In contrast, significant oscillation along the dominant direction emerges for the misaligned regions (see red boxes). Figure 4b directly correlates this instability with sensitivity to delay. We observe that the slowdown ratio is minimized near basis-aligned regions, indicating that delay has a negligible impact on convergence (see green dotted lines). In contrast, this ratio increases in misaligned regions where delay substantially impedes the convergence (see red dotted lines). These results support that basis misalignment renders coordinate-wise adaptive scheme of Adam-type optimizers susceptible to delay-induced convergence instability.

### 2.3. Theoretical Verification

Beyond the empirical observations, we provide a formal convergence analysis to demonstrate that Adam suffers significantly from delayed gradients under basis misalignment. Specifically, we prove that the negative effect of delay $\tau$ on the convergence rate is exacerbated for the basis-misaligned case compared to the aligned case. We first present assump-

tions necessary for our analysis.

**Assumption 2.1.** (Coordinate-wise bounded noise) For each coordinate $i \in \{1, \cdots, d\}$, there exists $\sigma_i$ such that $\mathbb{E}_\xi[\nabla_i f(w; \xi) - \mathbb{E}_\xi[\nabla_i f(w; \xi)]]^2 \leq \sigma_i^2$ for all $w \in \mathbb{R}^d$ where $\xi$ denotes the stochasticity of data.

**Assumption 2.2.** ($c$-coordinate-wise $\ell_\infty$ smoothness) $f$ is $c$-smooth coordinate-wisely w.r.t. $\ell_\infty$ norm with $c = (C_1, \cdots, C_d) \in \mathbb{R}^d$, *i.e.*, for each coordinate $i \in \{1, \cdots, d\}, |\nabla_i f(w) - \nabla_i f(w')| \leq C_i \|w - w'\|_\infty$ for any $w, w' \in \mathbb{R}^d$. This implies that $f$ is $C$-smooth w.r.t. $\ell_\infty$ norm where $C = \sum_{i=1}^d C_i$, *i.e.*, $\|\nabla f(w) - \nabla f(w')\|_1 \leq C\|w - w'\|_\infty$ for all $w, w' \in \mathbb{R}^d$.

These coordinate-wise assumptions enable capturing Adam's coordinate-wise adaptivity in the convergence rate. Specifically, $C$ locally corresponds to the $(1, 1)$-norm of the Hessian matrix, defined as $\|\nabla^2 f(w)\|_{1,1} := \sum_{i,j} |(\nabla^2 f(w))_{i,j}|$. For a fixed eigenvalue spectrum, this norm is minimized when the Hessian is diagonal and increases under basis misalignment. Thus, it acts as a promising proxy for basis misalignment (Xie et al., 2025; Zhang et al., 2025a). We are now ready to present the convergence theorem for asynchronous Adam under delay.

**Theorem 2.3.** *Let $f$ be a non-convex function where Assumptions 2.1 and 2.2 hold. Define $\Delta_0 \triangleq (f(w_0) - \min_w f(w))$ as the initial suboptimality gap. Assume that initial second moment $v_0$ and step size $\eta$ satisfy $v_0 + \epsilon > (\sum_{i=1}^d \sigma_i^2 + \|\nabla f(w_0)\|_\infty^2 + \sum_{i=1}^d C_i^2 \eta^2)/\texttt{poly}(T)$. Then, with appropriate choices of $\eta$ and $\beta_2$, the convergence rate of asynchronous Adam with $\beta_1 = 0$ under delay $\tau$ is as follows:*

$$\min_{\frac{T}{2} < t \leq T} \mathbb{E}\|\nabla f(w_t)\|_1$$

$$= \mathcal{O}\left( \sqrt{\frac{(1 + d\tau)\Delta_0 C}{T}} + \sqrt{\sum_{i=1}^d \sigma_i \left( \frac{(1 + d\tau)\Delta_0 C}{T} \right)^{1/4}} \right.$$

$$\left. + \sum_{i=1}^d \sigma_i \left( \frac{\log T}{T} \right)^{1/4} \right).$$

The proof is presented in Appendix E.

This result reveals that the impact of delay $\tau$ is tightly coupled with basis misalignment $C$ as $\tau$ enters the bound multiplicatively with $C$. Specifically, for a fixed delay $\tau$, the relative contribution of the delay-dependent terms to the total bound increases with $C$, thereby exacerbating the penalty for basis misalignment. Thus, while the effects of delay are suppressed for basis-aligned cases (*i.e.*, small $C$), they are aggressively amplified for misaligned cases (*i.e.*, large $C$) where the delay-dependent terms begin to dominate the bound. This formalizes our empirical observation that the impact of delay increases when the Hessian eigenbasis is misaligned with the standard basis.

**Remark** The result indicates that delay $\tau$ slows down the deterministic convergence rate from $\mathcal{O}(\sqrt{1/T})$ to $\mathcal{O}(\sqrt{\tau/T})$, which aligns with the analysis from the asynchronous optimization literature (Stich & Karimireddy, 2020; Koloskova et al., 2022). The additional dimension factor $d$ and the $\tau$ dependence appearing in the stochastic term stem from the coordinate-wise adaptive denominators of Adam. Also, when $\tau = 0$, the result recovers the convergence rate of Adam under $\ell_\infty$ smoothness without delay (Xie et al., 2025). The extension to $\beta_1 > 0$ is given in Appendix E.2.

**Extension to Stage-Dependent Delay** Beyond the uniform delay setting, we extend the analysis to the stage-dependent delay setting where each stage incurs a distinct delay, to provide a more precise characterization of the convergence behavior in practical asynchronous pipeline parallelism. Let $\mathcal{S}_1, \ldots, \mathcal{S}_K$ denote the partition of parameter coordinates across $K$ pipeline stages, where coordinate $i \in \mathcal{S}_k$ incurs a delay $\tau_i = K - k$. The bound in Theorem 2.3 then carries over with $\tau$ replaced by a tighter, stage-aware effective delay:

$$\tau' \triangleq \sqrt{\frac{\sum_{i=1}^d C_i^2 \tau_i^2}{\sum_{i=1}^d C_i^2}} = \sqrt{\frac{\sum_{k=1}^K (K-k)^2 \sum_{i \in \mathcal{S}_k} C_i^2}{\sum_{i=1}^d C_i^2}}. \quad (3)$$

This formulation reveals that earlier pipeline stages, where the $\tau_i$ is most severe, have a larger impact on $\tau'$ and thus dominate the convergence degradation. The formal statement and proof are provided as Theorem E.6 in Appendix E.1.

## 3. Mitigating the Impact of Delay

Our previous analysis suggests that the negative effects of delay can be mitigated in a basis-aligned space. We propose to achieve this through basis rotation which transforms the optimization space to align the Hessian eigenbasis with the standard basis.

### 3.1. Basis Rotation

Standard Adam update

$$w_t = w_{t-1} - \eta_{t-1} \frac{\text{EMA}(\nabla f(w_{t-1}))}{\sqrt{\text{EMA}(\nabla f(w_{t-1})^2) + \epsilon}} \quad (4)$$

relies on coordinate-wise scaling. As demonstrated in Section 2, this scaling becomes ineffective when the Hessian eigenbasis is misaligned with the standard basis, causing delayed gradients to introduce significant noise into the optimization process. The core idea of basis rotation is to mitigate the impact of delay by taking an optimization step

---

**Algorithm 1** Adam with Basis Rotation

---
1: **for** $t = 1, 2, \ldots, T$ **do**
2:      Sample batch $B_t$
3:      $G_t \leftarrow \nabla f_W(W_{t-1}; B_t) \in \mathbb{R}^{m \times n}$
4:      $M_t \leftarrow \beta_1 M_{t-1} + (1 - \beta_1) G_t$
5:      **if** $t \bmod \text{freq} = 0$ **then**
6:          $U, V \leftarrow \texttt{Eigenbasis-Estimation}(G_t, M_t, U, V)$
7:      **end if**
8:      $\widetilde{G}_t \leftarrow U^\top G_t V$
9:      $\widetilde{M}_t \leftarrow U^\top M_t V$
10:     $\widetilde{V}_t \leftarrow \beta_2 \widetilde{V}_{t-1} + (1 - \beta_2) \widetilde{G}_t \odot \widetilde{G}_t$
11:     $W_t \leftarrow W_{t-1} - \eta_t U \left( \frac{1}{\sqrt{\widetilde{V}_t} + \epsilon} \odot \widetilde{M}_t \right) V^\top$
12: **end for**

---

in the basis-aligned coordinate system $\tilde{w} = \mathcal{U}^\top w$:

$$\tilde{w}_t = \tilde{w}_{t-1} - \eta_{t-1} \frac{\text{EMA}(\mathcal{U}^\top \nabla f(\mathcal{U} \tilde{w}_{t-1}))}{\sqrt{\text{EMA}((\mathcal{U}^\top \nabla f(\mathcal{U} \tilde{w}_{t-1}))^2) + \epsilon}}, \quad (5)$$

where $\mathcal{U}$ is a rotation matrix whose columns are estimated eigenvectors of the Hessian. Projecting the update back onto the original space gives

$$w_t = w_{t-1} - \eta_{t-1} \mathcal{U} \frac{\text{EMA}(\mathcal{U}^\top \nabla f(w_{t-1}))}{\sqrt{\text{EMA}((\mathcal{U}^\top \nabla f(w_{t-1}))^2) + \epsilon}}. \quad (6)$$

Crucially, when $\mathcal{U}$ perfectly aligns with the Hessian eigenbasis, the Hessian eigenbasis in the rotated space matches exactly with the standard coordinate basis. By ensuring that adaptive scaling is applied in the basis-aligned space, basis rotation restores the efficacy of curvature-aware adaptivity and mitigates the adverse effects of gradient delay. Detailed derivation of Equation (6) is given in Appendix C.

The full procedure of basis rotation for a weight matrix $W \in \mathbb{R}^{m \times n}$ is detailed in Algorithm 1. To mitigate the prohibitive computational and memory costs for computing $\mathcal{U}$ in Equation (6), basis rotation adopts two structural assumptions on the Hessian: (i) block-diagonality and (ii) Kronecker factorization. The former assumes a block-diagonal Hessian, enabling matrix-wise rotation rather than rotating the entire parameter space at once. This assumption is empirically supported in Transformer (Zhang et al., 2024; 2025b; Abreu et al., 2026). The latter allows the rotation matrix $\mathcal{U}_W \in \mathbb{R}^{mn \times mn}$ to be factorized into $U \in \mathbb{R}^{m \times m}$ and $V \in \mathbb{R}^{n \times n}$, making the rotation computationally tractable for large models (Martens & Grosse, 2015; Gupta et al., 2018; Vyas et al., 2025). Together with infrequent basis updates, basis rotation remains efficient at scale.

### 3.2. Eigenbasis Estimation

We now introduce efficient $\texttt{eigenbasis-estimation}$ strategies which theoretically induce basis alignment under

---

**Algorithm 2** `Eigenbasis-Estimation`

---

**Require:** Approximation source $\mathcal{S}$, Rotation geometry $\mathcal{G}$,
    $G_t, M_t, U, V$

1: **if** $\mathcal{S} = 2^{\text{nd}}$ **then**
2:     $L \leftarrow \beta_2 L + (1 - \beta_2) G_t G_t^\top$
3:     $U \leftarrow \text{Power}(L, U)$
4:     **if** $\mathcal{G} = \text{Bilateral}$ **then**
5:         $R \leftarrow \beta_2 R + (1 - \beta_2) G_t^\top G_t$
6:         $V \leftarrow \text{Power}(R, V)$
7:     **end if**
8:     **if** $\mathcal{G} = \text{Unilateral}$ **then**
9:         $V \leftarrow I$
10:     **end if**
11: **end if**
12: **if** $\mathcal{S} = 1^{\text{st}}$ **then**
13:     $U \leftarrow \text{Power}(M_t M_t^\top, U)$
14:     **if** $\mathcal{G} = \text{Bilateral}$ **then**
15:         $V \leftarrow \text{Power}(M_t^\top M_t, V)$
16:     **end if**
17:     **if** $\mathcal{G} = \text{Unilateral}$ **then**
18:         $V \leftarrow I$
19:     **end if**
20: **end if**

---

the assumptions of Section 3.1. These strategies are categorized along two design axes: approximation source ($\mathcal{S}$) and rotation geometry ($\mathcal{G}$). This taxonomy provides a unified design space for balancing estimation fidelity against memory efficiency, depending on the demands and constraints of the training environment. A detailed memory overhead analysis of each strategy is provided in Appendix H.

The first axis $\mathcal{S}$ determines the statistical source used to approximate the Hessian. The high-fidelity version ($2^{\text{nd}}$) utilizes the second moments $L = \mathbb{E}[GG^\top]$ and $R = \mathbb{E}[G^\top G]$ to approximate the empirical Fisher, where $G$ is the gradient matrix. This quantity has frequently been used as a computationally efficient proxy for the Hessian (Roux et al., 2007; Singh & Alistarh, 2020; Frantar et al., 2021). In contrast, $\mathcal{S} = 1^{\text{st}}$ variant employs first-order moments ($\mathbb{E}[GG^\top] \approx \mathbb{E}[G]\mathbb{E}[G]^\top$), eliminating dedicated storage for $L$ and $R$ by leveraging the existing momentum buffer.

The second axis $\mathcal{G}$ determines the rotation geometry used to diagonalize the approximating matrix. The bilateral version applies a two-sided rotation to capture the full Kronecker-factored structure, while the unilateral variant applies a one-sided rotation to the smaller dimension of the gradient. This reduces overhead by storing eigenvectors of either $L$ or $R$, which is particularly beneficial for rectangular matrices.

These strategies can theoretically reduce the $(1, 1)$-norm of the Hessian, which is a proxy for the degree of misalignment as introduced in Section 2.3. Define the rotated

Hessians $H_U$ and $H_{U,V}$ corresponding to the transformations $\widetilde{W} = U^\top W$ (unilateral) and $\widetilde{W} = U^\top W V$ (bilateral) respectively. Then, the following theorem holds.

**Theorem 3.1.** *Let $U$ and $V$ be the matrices whose columns are eigenvectors of $\mathbb{E}[GG^\top]$ and $\mathbb{E}[G^\top G]$ respectively. If Hessian admits a Kronecker-factorized empirical Fisher, the following inequalities hold:*

$$\|H_{U,V}\|_{(1,1)} \leq \|H_U\|_{(1,1)} \leq \|H\|_{(1,1)}.$$

*Moreover, $\|H_{U,V}\|_{(1,1)}$ achieves the global minimum over all rotations.*

The theorem shows that the bilateral rotation minimizes basis misalignment while both versions provide better alignment than the original space. The proof and further result for $\mathcal{S} = 1^{\text{st}}$ are provided in Appendix F.

The full procedure covering both axes is presented in Algorithm 2. We use a single step of power iteration followed by QR decomposition to efficiently compute eigenvectors (Wang et al., 2024). While our `eigenbasis-estimation` framework shares similarities with recent optimizers—such as SOAP (Vyas et al. (2025); $\mathcal{S} = 2^{\text{nd}}$, $\mathcal{G} = $ bilateral) and the full-rank version of GaLore (Zhao et al. (2024); $\mathcal{S} = 1^{\text{st}}$, $\mathcal{G} = $ unilateral)—we distinguish our approach by unifying these strategies into a single framework rather than adopting individual optimizers. This controlled setup allows us to isolate the effects of Hessian geometry from other implementation variables. See Appendix G for a detailed comparison.

## 4. Experiments

### 4.1. Experimental Setup

We evaluate the performance of basis rotation on the language modeling task with a standard decoder-only Transformer (Vaswani et al., 2017) ranging from 95M to 1B parameters. The models are trained on 1B tokens randomly selected from OpenWebText (Gokaslan & Cohen, 2019). We evaluate basis rotation against three baselines which are primary strategies for asynchronous pipeline parallelism in the literature: (1) PipeDream (Narayanan et al., 2019) which serves as the vanilla baseline by not explicitly addressing delay, (2) PipeDream-LR (Yang et al., 2021) which schedules the stage-wise learning rate depending on the level of delay, and (3) Nesterov (Ajanthan et al., 2025) which incorporates Nesterov momentum to address delayed gradients. We use $\mathcal{S} = 2^{\text{nd}}$, $\mathcal{G} = $ bilateral strategy for `eigenbasis-estimation` and set the basis update frequency to 10 unless specified otherwise. Finally, we employ weight stashing (Narayanan et al., 2019) across all methods to ensure consistency between the weights used in the forward and backward passes. Experimental details can be found in Appendix D.2.

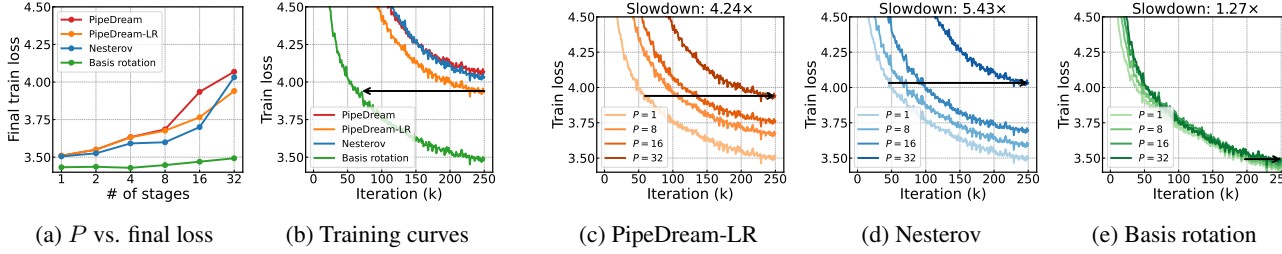

(a) $P$ vs. final loss     (b) Training curves     (c) PipeDream-LR     (d) Nesterov     (e) Basis rotation

*Figure 5.* Performance of different methods when increasing the number of stages $P$ for the same model (with 95M parameters). (a) Basis rotation maintains stable performance while baselines suffer significantly from delay. (b) Basis rotation shows much faster convergence under large delay compared to baselines (for $P = 32$). (c-e) Basis rotation reduces slowdown, *i.e.*, the iteration ratio required to reach target loss for $P = 32$ relative to $P = 1$, by a large margin compared to baselines. More results including other baselines and different settings are presented in Figures 12, 13, 18, 19 and 21 and Table 3 of Appendix I.

## 4.2. Main Results

We first evaluate the robustness of basis rotation against delay by increasing the number of pipeline stages $P$ for the same model. The results summarized in Figure 5a demonstrate that basis rotation consistently outperforms baselines with the performance gap widening significantly as $P$ increases. While prior work has demonstrated the benefits of optimization in the Hessian eigenbasis under the standard zero-delay regime (Vyas et al., 2025; Eschenhagen et al., 2025), our results indicate that these advantages are substantially amplified in the presence of large delays. Notably, Figure 5b demonstrates that at $P = 32$, basis rotation achieves the same training loss with 71.6% fewer iterations than the best-performing baseline. To quantify robustness to delay, the right three panels of Figure 5 report the slowdown of each method, defined as the ratio of the number of iterations required to reach a fixed loss threshold at $P = 32$ relative to $P = 1$. As shown in Figures 5c and 5d, even the best-performing baseline shows a large slowdown of $4.24\times$. In contrast, basis rotation in Figure 5e exhibits substantially improved robustness to delay with only $1.27\times$ slowdown.

Next, we investigate the scalability of basis rotation by jointly increasing the number of Transformer blocks and the number of pipeline stages $P$, assigning one block to each stage. This setup reflects the practical necessity of using deeper pipelines to accommodate the memory footprint of larger models. Figures 6a and 6b reveal a critical failure in baseline methods; increasing model size leads to higher training loss, directly contradicting standard scaling laws (Kaplan et al., 2020; Hoffmann et al., 2022). In contrast, basis rotation in Figure 6c successfully restores scalability, achieving performance improvements as model size grows.

Finally, we evaluate basis rotation on a 1B parameter model to verify its efficacy at a larger scale. Specifically, we set $P = 24$ and increase the number of embedding dimensions to scale the model. Figure 7 demonstrates the performance gap between basis rotation and the baselines widens for larger models. For 1B model, basis rotation achieves the

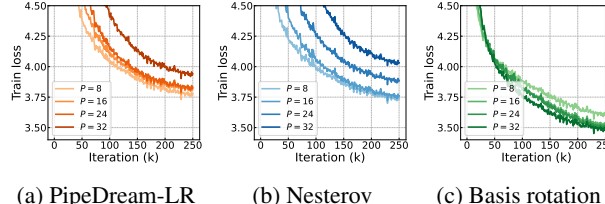

(a) PipeDream-LR    (b) Nesterov    (c) Basis rotation

*Figure 6.* Performance of different methods when increasing $P$ by scaling the number of blocks. While scaling the model leads to increased loss for baselines (a,b), it decreases the loss for basis rotation (c). More results are presented in Figure 14 of Appendix I.

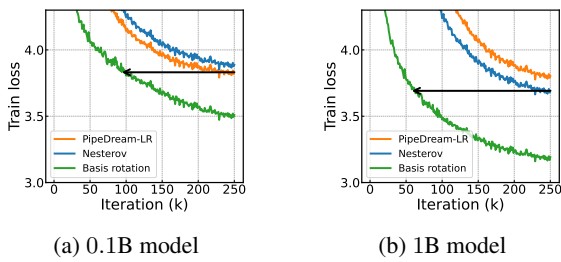

(a) 0.1B model      (b) 1B model

*Figure 7.* Performance of different methods at $P = 24$ for (a) 0.1B and (b) 1B models. The gap between basis rotation and baselines increases further for larger models.

same training loss with 76.8% fewer iterations than the best-performing baseline, surpassing the 62.4% reduction in smaller models. This advantage further increases at 3B scale, where basis rotation reduces iterations by 81.7% (see Figure 20 in Appendix I). The results show that the advantages of our approach scale positively with model size.

Overall, our results underscore that the performance degradation in asynchronous training is not an unavoidable cost of staleness but rather a byproduct of optimizer sensitivity to delay. Basis rotation effectively eliminates this bottleneck, unlocking a new regime for bubble-free execution at scale.

## 4.3. More Results

**Basis Approximation Sensitivity** To evaluate how the precision of the eigenbasis approximation af-

fects robustness to delay, we compare different `eigenbasis-estimation` strategies in Figure 8. As seen in Table 8a, basis rotation exhibits higher robustness with a closer approximation to the Hessian eigenbasis. Specifically, 2nd results in a smaller slowdown than 1st while bilateral outperforms unilateral. The results confirm that basis misalignment is the primary factor of the performance degradation under delay and a more accurate basis rotation effectively mitigates the issue. We also note that even the least accurate strategy (1st / unilateral) outperforms the best-performing baseline by a large margin (see Figure 8b). The results suggest that our approach is highly effective even in resource-constrained settings where more sophisticated estimation strategies may be computationally expensive.

**Computational Efficiency**  To address potential concerns regarding the computational cost of the basis update, we compare its wall-clock efficiency against baselines. Figure 9a demonstrates that basis rotation remains superior in terms of GPU hours, reaching the same training loss with 54.3% less amount of time than the most competitive baselines. Furthermore, this overhead can be reduced by adjusting the basis update frequency with a marginal performance degradation. As shown in Figure 9b, basis rotation remains significantly more efficient than the baselines even at an update frequency of 100 iterations.

**Stage-aware Basis Rotation**  Beyond uniform update frequencies, we investigate a stage-aware strategy (detailed in Appendix I) that allocates the computational budget for subspace updates in proportion to the gradient delay at each stage. Figure 9c shows this achieves a 29.2% speedup in convergence compared to the uniform-frequency baseline, while preserving the same total computational budget. This empirical gain aligns with the theoretical insight from Equation (3): the effective delay $\tau'$ is dominated by the misalignment magnitude ($C_i^2$) at the earliest pipeline stages, where the per-stage delay ($K - k$) is most severe. Suppressing it thus reduces $\tau'$ and accelerates convergence. An ablation with inversely-ordered frequency allocation in Figure 17 of Appendix I confirms this: the reversed strategy degrades performance relative to the uniform baseline.

**Robustness without Weight Stashing**  The weight stashing employed throughout our main experiments ensures correct backpropagation by storing the weights used for the forward pass. However, it incurs memory overhead that scales linearly with the number of pipeline stages, making it prohibitive in memory-constrained environments. To evaluate whether the benefits of basis rotation persist in such resource-limited settings, we omit weight stashing, resulting in incorrect gradient computation (Gaunt et al., 2017; Huo et al., 2018). Figure 10a shows that basis rotation

| Method | Estimation $\mathcal{S}$ | $\mathcal{G}$ | Slowdown |
|---|---|---|---|
| PipeDream-LR | — | — | $4.24\times$ |
| Basis rotation | 1st | Uni | $2.55\times$ |
|  |  | Bi | $1.77\times$ |
|  | 2nd | Uni | $1.66\times$ |
|  |  | Bi | $1.27\times$ |

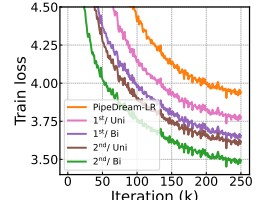

(a) Convergence slowdown  (b) Training loss

*Figure 8.* Comparison of different `eigenbasis-estimation` strategies. (a) High-fidelity estimation leads to a smaller slowdown under delay. (b) High-fidelity estimation achieves faster convergence for $P = 32$. Even the least accurate estimation (1st/unilateral) outperforms the best-performing baseline. More results are presented in Figure 16 of Appendix I.

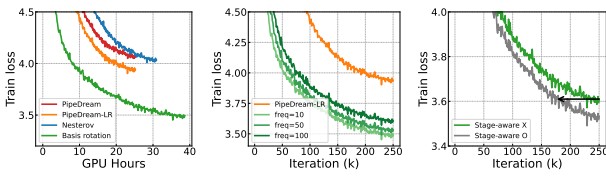

(a) GPU Hours (b) Update frequency (c) Stage-aware BR

*Figure 9.* Efficiency of basis rotation. (a) Basis rotation outperforms other methods by a large margin in terms of GPU hours. (b) Basis rotation shows little performance degradation with infrequent basis updates. (c) Stage-aware basis rotation achieves superior convergence compared to the uniform update frequency under the same total computational budget.

maintains a significant performance advantage over baselines: it remains robust without weight stashing (Figure 10c), whereas the best-performing baseline exhibit severe degradation (Figure 10b). Additional experiments with PipeMare-style weight prediction (Yang et al., 2021)—which partially addresses the weight discrepancy problem—likewise show basis rotation outperforming baselines (Figure 15 in Appendix I), confirming that our approach is significantly more resilient to incorrect backpropagation.

**Empirical Validation of Basis Alignment**  To confirm whether our core arguments made in Section 2 hold in practical LLM pre-training, we empirically validate that basis alignment reduces update oscillations along dominant eigenvectors and decreases the Hessian $(1, 1)$-norm. We estimate the dominant and non-dominant eigenvectors midway through training and track parameter updates along them. As shown in Figure 11, standard training exhibits severe oscillations along the dominant eigenvector, whereas basis rotation substantially dampens them; non-dominant directions remain stable in both cases. We further measure the Hessian $(1, 1)$-norm – our theoretical proxy for basis misalignment – via trace estimation with random Cauchy vectors (Xie et al., 2025). Basis rotation reduces the normalized Hessian $(1, 1)$-norm (per parameters) from 0.5436

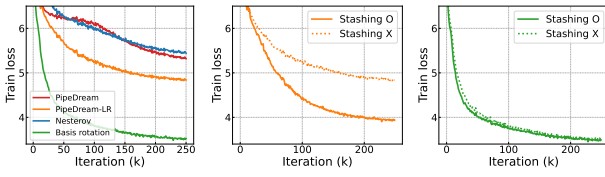

(a) W/o stashing     (b) PipeDream-LR     (c) Basis rotation

*Figure 10.* Performance of different methods without weight stashing for $P = 32$. (a) Basis rotation outperforms baselines with a large margin. (b,c) While baselines exhibit severe degradation without weight stashing, basis rotation remains robust.

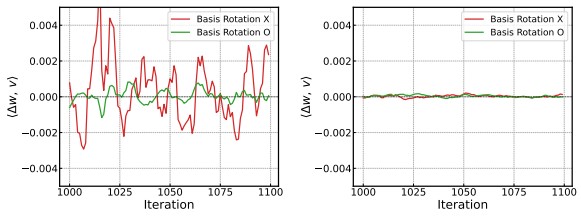

(a) Dominant eigenvector     (b) Nondominant eigenvector

*Figure 11.* Oscillation of parameter updates along the (a) dominant and (b) non-dominant eigenvectors of the Hessian. Standard training without basis rotation exhibits severe oscillations along the dominant direction. Applying basis rotation effectively dampens these oscillations. Updates along the non-dominant direction remain stable in both settings.

to 0.1228. This confirms that our framework realigns the optimization space, neutralizing the conditions that make adaptive optimizers vulnerable to gradient delay (see Appendix D.3 for detailed experimental settings).

**Additional Results** We further compare basis rotation against preconditioned optimizers (SOAP (Vyas et al., 2025), Muon (Jordan et al., 2024), Scion (Pethick et al., 2025)) and verify generalization to Mixture-of-Experts (MoE) architectures; see Appendix I for details.

## 5. Related Works

**Pipeline Parallelism** GPipe (Huang et al., 2019) introduces synchronous pipeline parallelism but its efficiency is limited by pipeline bubbles. While several works propose advanced pipeline scheduling algorithms to improve device utilization, they still suffer from synchronization bottlenecks (Fan et al., 2021; Li & Hoefler, 2021). Asynchronous pipeline parallelism significantly maximizes throughput by entirely removing the synchronization step but exhibits performance degradation due to delayed gradients (Narayanan et al., 2019; 2021). Prior works address the issue with learning rate scheduling (Yang et al., 2021; Zhuang et al., 2021), Nesterov momentum (Ajanthan et al., 2025), and future weight prediction (Chen et al., 2018; Guan et al., 2019). However, these methods fail to effectively address delayed

gradients under large-scale settings.

**Adam for LLM Training** Adam (Kingma & Ba, 2015) is widely used for training LLMs due to its superior performance over SGD. Recent works explain this success from various aspects including directional smoothness (Zhang et al., 2020; Pan & Li, 2022), heterogeneity of the Hessian spectrum in Transformers (Zhang et al., 2024), and heavy-tailed noise of language data (Kunstner et al., 2024). Meanwhile, Adam loses its advantage over SGD when the Hessian is not near-diagonal, or equivalently, when the Hessian eigenbasis is not aligned to the standard basis (Wang & Wiens, 2020; Zhang et al., 2025b). This is because Adam is not equivariant under rotation and its coordinate-wise adaptivity fails to work effectively for the basis-misaligned loss landscapes (Xie et al., 2025; Zhang et al., 2025a). We demonstrate that this problem exacerbates the impact of delay.

**Hessian Approximation** Hessian is often approximated via Gauss-Newton matrix (Schraudolph, 2002; Botev et al., 2017), Fisher (Amari, 1998; Martens, 2020), or empirical Fisher (Roux et al., 2007; Singh & Alistarh, 2020; Frantar et al., 2021), and their Kronecker-factored variants (Martens & Grosse, 2015; Gupta et al., 2018; Vyas et al., 2025). While Gauss-Newton and Fisher matrices serve as a robust proxy for the Hessian (Martens, 2020; Grosse, 2021), they require multiple backpropagation which leads to complex scheduling in asynchronous pipeline parallelism. Recent works show that Kronecker-factored empirical Fisher can serve as a practical surrogate for the Gauss-Newton matrix (Morwani et al., 2025). Based on these works, we utilize the eigenvectors of the Kronecker-factored empirical Fisher to perform basis rotation, ensuring a principled yet computationally tractable transformation.

## 6. Conclusion

In this work, we show that asynchronous pipeline parallelism is fundamentally challenged by gradient staleness: its convergence properties and model performance degrade critically as the delay grows with pipeline depth or model size. We identify that the root of this degradation lies in basis misalignment, which prevents a coordinate-wise adaptive optimizer from effectively navigating the loss landscape and leads to amplified oscillations under delay. To address this, we propose basis rotation, a framework that rotates the optimization space to realign the Hessian eigenbasis with the standard coordinate basis. Our extensive experiments show that basis rotation effectively neutralizes the impact of delay, significantly accelerating convergence in practical settings, and restoring scalable asynchronous training.

## Acknowledgements

We thank Dongyeop Lee, Jinseok Chung, and Jihun Kim for helpful discussions during the early development of this project. This work was supported by the Institute of Information & Communications Technology Planning & Evaluation (IITP) grant funded by the Korean government (MSIT) (RS-2019-II191906, Artificial Intelligence Graduate School Program (POSTECH), RS-2026-25507427, Development of Efficient Architectures and Training Techniques for High-Performance Lightweight AI Models), the National Research Foundation of Korea (NRF) grant funded by the Korea government (MSIT) (RS-2023-00210466), and Korea Basic Science Institute (National research Facilities and Equipment Center) grant funded by the Korea government(MSIT) (RS-2026-25500419). Hyunji Jung was supported by the Korea Student Aid Foundation (KOSAF). Sungbin Shin was supported by Kwanjeong Educational Foundation Scholarship.

## Impact Statement

This paper presents work aimed at advancing the field of machine learning by proposing basis rotation to mitigate delay in asynchronous pipeline parallelism. While the continued development of machine learning models has broad societal implications, there are no specific ethical concerns or negative consequences of this research that we feel must be highlighted here.

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

# Contents

# A. Analysis for the Number of Stages for Pipeline Parallelism

This analysis analyzes the number of stages for pipeline parallelism when training LLMs of varying sizes. We calculate the minimum number of pipeline stages required to fit different model architectures onto standard GPU devices with different memory capacities. By modeling the memory footprint of a single Transformer block including parameters, optimizer states, and activation, we derive a formula to determine how many blocks can fit on a single device, and consequently, how many devices (stages) are needed for the full model.

**Setting** We assume a standard training recipe commonly used for dense LLMs. Specifically, we assume mixed precision training with AdamW optimizer for a standard Transformer architecture. We assume that gradient checkpointing is applied at the beginning of each Transformer block while weight stashes are offloaded to CPU to save memory. We only employ pipeline parallelism and do not use context, tensor, or fully sharded data parallelism. This is because these strategies require frequent collective communications and can be impractical for low-bandwidth settings (Rajbhandari et al., 2020; Shoeybi et al., 2019).

**Notation** We denote the embedding dimension, the number of attention heads, and the sequence length as $h$, $a$, and $s$ respectively. We also denote batch size as $b$. $W$ represents the number of parameters in a single Transformer block. $L$ represents the total number of Transformer blocks.

**Memory required for a single block** To determine the number of stages, we first calculate the memory required for a single Transformer block. The memory consumption for one Transformer block consists of parameters, gradients, optimizer states, and activations. Assuming mixed-precision training, we need the following memory: (1) $2W$ bytes for half-precision parameters, (2) $2W$ bytes for half-precision gradients, (3) $4W + 4W + 4W = 12W$ bytes for full-precision optimizer states (each corresponding to full-precision master weights, first momentum, and second momentum), (4) $\approx 34sbh + 5bas^2$ bytes for activations. The memory for activation is the result from Korthikanti et al. (2023). Summing these components, the total memory required for a single block, $M_{\text{block}}$, is

$$M_{\text{block}} = 16W + 34sbh + 5bas^2 \quad \text{(bytes)} \tag{7}$$

Consequently, the total memory for $N$ blocks is $N \times M_{\text{block}}$.

**Calculating Required Stages** Let $m$ be the memory capacity of a single device (in bytes). We calculate $N_{\max}$, the maximum number of Transformer blocks that can fit on a single device, using the inequality:

$$N M_{\text{block}} \leq m, \tag{8}$$

which leads to $N_{\max} = \left\lfloor \frac{m}{16W + 34sbh + 5bas^2} \right\rfloor$. Based on $N_{\max}$, the minimum number of pipeline stages $P$ required to host the full model (which has $L$ total blocks) is determined as follows: (1) when $N_{\max} \geq 1$, at least one block fits on a device and $P = \left\lceil \frac{L}{N_{\max}} \right\rceil$ [3], and (2) when $N_{\max} = 0$, a single device cannot hold even one Transformer block and $P \geq 2L$ (since we require at least two stages to allocate a single block).

We calculate the required number of stages ($P$) for various LLaMA models (Touvron et al., 2023a; Dubey et al., 2024) across different GPU hardware. Specifically, we assume a batch size of 1 for simplicity, *i.e.*, $b = 1$. We also assume a sequence length of 4096, *i.e.*, $s = 4096$, which are typically used for initial-phase training of recent LLMs (Touvron et al., 2023b; Team et al., 2024; Gunter et al., 2024; Abdin et al., 2024). We note that recent LLMs typically use much larger values of $b$ and $s$ so the number of requires stages may increase further.

---

[3]This is when we ignore the memory needed for loading embedding layers and language model head. The number of stages may further increase for large embedding layers.

*Table 1.* Number of required stages ($P$) for different LLaMA models with sequence length $s = 4096$ and batch size $b = 1$. Values marked with * indicate that a single Transformer block cannot fit on one device ($N_{max} = 0$).

| Model | $h$ | $a$ | $W$ | $L$ | RTX3070 (8GB) | RTX3080 (16GB) | RTX3090 (24GB) | A6000 (48GB) | A100 (80GB) |
|---|---|---|---|---|---|---|---|---|---|
| Llama 3.2 1B | 2048 | 32 | $\approx 67M$ | 16 | 16 | 6 | 4 | 2 | 1 |
| Llama 3.2 3B | 3072 | 24 | $\approx 113M$ | 28 | 28 | 10 | 6 | 3 | 2 |
| LLaMA 1-7B | 4096 | 32 | $\approx 202M$ | 32 | 32 | 16 | 11 | 5 | 3 |
| LLaMA 1-13B | 5120 | 40 | $\approx 317M$ | 40 | $\geq 80^*$ | 40 | 20 | 8 | 5 |
| LLaMA 1-33B | 6656 | 52 | $\approx 535M$ | 60 | $\geq 120^*$ | 60 | 60 | 20 | 12 |
| LLaMA 1-65B | 8192 | 64 | $\approx 810M$ | 80 | $\geq 160^*$ | $\geq 160^*$ | 80 | 40 | 20 |
| Llama 3.1 405B | 16384 | 128 | $\approx 3.19B$ | 126 | $\geq 512^*$ | $\geq 512^*$ | $\geq 512^*$ | $\geq 512^*$ | 126 |

The results are presented in Table 1. For devices with small memory (*e.g.*, RTX3070), we need at least 80 stages even for 13B models. For larger models, the number of stages easily increases to hundreds. Even with a device with larger memory (*e.g.*, A6000), we need tens of stages for moderate-size models (*e.g.*, 33B, 65B). This analysis confirms that the number of stages in pipeline parallelism can easily reach tens or hundreds for the case of LLMs.

## B. Stability of the Linearized Trajectory under Delay (Section 2.1)

Before delving into the main argument, we first define the standard and delayed trajectories. The actual trajectory without delay is updated by

$$x_t = x_{t-1} + u_t = x_{t-1} - \eta \frac{m_t}{\sqrt{v_t + \epsilon}}, \tag{9}$$

where $m_t = \beta_1 m_{t-1} + (1 - \beta_1)g_t$ and $v_t = \beta_2 v_{t-1} + (1 - \beta_2)g_t^2$, and $g_t = \nabla f(x_{t-1}; \xi_t)$ denotes the stochastic gradient evaluated at the current iterate $x_{t-1}$ with batch $\xi_t$. The trajectory generated by the delayed gradients obeys the update rule

$$\tilde{x}_t = \tilde{x}_{t-1} + \tilde{u}_t = \tilde{x}_{t-1} - \eta \frac{\tilde{m}_t}{\sqrt{\tilde{v}_t + \epsilon}}, \tag{10}$$

with the moments following the identical recursion

$$\tilde{m}_t = \beta_1 \tilde{m}_{t-1} + (1 - \beta_1)\tilde{g}_t, \qquad \tilde{v}_t = \beta_2 \tilde{v}_{t-1} + (1 - \beta_2)\tilde{g}_t^2. \tag{11}$$

The only difference from the non-delayed trajectory lies in which gradient is consumed. Under a delay of $\tau$, the gradient applied at update step $t$ was produced $\tau$ updates earlier, i.e., evaluated at the iterate $\tilde{x}_{t-1-\tau}$:

$$\tilde{g}_t = \begin{cases} \nabla f(\tilde{x}_0; \xi_t), & t \leq \tau, \\ \nabla f(\tilde{x}_{t-1-\tau}; \xi_t), & t > \tau. \end{cases} \tag{12}$$

Now, we can derive the following statement: *When the gradient signal dominates the noise ($\nabla f(x; \xi) \approx \nabla f(x)$), the linearized non-delayed trajectory ($u_t \approx u$ for all $1 \leq t \leq s$) makes the delayed trajectory closely approximate the original ($\tilde{x}_t \approx x_t$ for all $0 \leq t \leq s$).*

We justify the statement by strong induction on $t$.

*Proof.* **Base case** ($t = 0$): $\tilde{x}_0 = x_0$ by initialization.

**Inductive step for $t \leq \tau$:** For every $j \leq t$, $\tilde{x}_0 = x_0$, hence by signal-dominance,

$$\tilde{g}_j = \nabla f(x_0; \xi_j) \approx \nabla f(x_0), \quad \tilde{m}_j \approx (1 - \beta_1)\nabla f(x_0), \quad \tilde{v}_j \approx (1 - \beta_2)\nabla f(x_0)^2. \tag{13}$$

Using the inductive hypothesis and the local consistency assumption ($u_t \approx u_1$):

$$\tilde{x}_t = \tilde{x}_{t-1} - \eta \frac{\tilde{m}_t}{\sqrt{\tilde{v}_t + \epsilon}}$$
$$\approx \tilde{x}_{t-1} - \eta \frac{(1-\beta_1)g_1}{\sqrt{(1-\beta_2)g_1^2 + \epsilon}}$$
$$= \tilde{x}_{t-1} + u_1$$
$$\approx x_{t-1} + u_t = x_t.$$

**Inductive step for $t > \tau$:** By the inductive hypothesis, $\tilde{x}_{t-1-\tau} \approx x_{t-1-\tau}$. Applying the signal-dominance assumption twice together with continuity:

$$\tilde{g}_t = \nabla f(\tilde{x}_{t-1-\tau}; \xi_t) \approx \nabla f(\tilde{x}_{t-1-\tau}) \approx \nabla f(x_{t-1-\tau}) \approx \nabla f(x_{t-1-\tau}; \xi_{t-\tau}) = g_{t-\tau}. \tag{14}$$

The cross-batch step $\xi_t$ vs. $\xi_{t-\tau}$ is bridged through the deterministic intermediate $\nabla f(x_{t-1-\tau})$, so no condition on the noise law is invoked. Consequently, the moving averages also approximate their un-delayed counterparts at step $t - \tau$, yielding $\tilde{m}_t \approx m_{t-\tau}$ and $\tilde{v}_t \approx v_{t-\tau}$. Therefore,

$$\tilde{x}_t = \tilde{x}_{t-1} - \eta \frac{\beta_1 \tilde{m}_{t-1} + (1-\beta_1)\tilde{g}_t}{\sqrt{\beta_2 \tilde{v}_{t-1} + (1-\beta_2)\tilde{g}_t^2 + \epsilon}}$$
$$\approx x_{t-1} - \eta \frac{\beta_1 m_{t-\tau-1} + (1-\beta_1)g_{t-\tau}}{\sqrt{\beta_2 v_{t-\tau-1} + (1-\beta_2)g_{t-\tau}^2 + \epsilon}}$$
$$= x_{t-1} + u_{t-\tau}.$$

Applying the local consistency assumption again ($u_{t-\tau} \approx u_t$), we conclude:

$$\tilde{x}_t \approx x_{t-1} + u_t = x_t.$$

Thus, by strong induction, the delayed trajectory $\tilde{x}_t$ follows the original trajectory $x_t$ closely along the optimization path. □

## C. Derivation of Adam's Equivalence under Fixed Rotation (Section 3.1)

In this section, we provide a detailed derivation to show that applying the standard Adam update in the basis-aligned coordinate system $\tilde{w} = \mathcal{U}^\top w$ mathematically recovers the basis rotation update rule in the original space (Equation (6)). Let the objective function in the rotated parameter space be defined as $\tilde{f}(\tilde{w}) \triangleq f(\mathcal{U}\tilde{w})$. By the multivariable chain rule, the gradient with respect to the rotated parameters $\tilde{w}$ is given by:

$$\nabla \tilde{f}(\tilde{w}_{t-1}) = \mathcal{U}^\top \nabla f(\mathcal{U}\tilde{w}_{t-1}) = \mathcal{U}^\top \nabla f(w_{t-1}).$$

Denoting $\tilde{g}_s = \nabla \tilde{f}(\tilde{w}_{s-1}) = \mathcal{U}^\top \nabla f(w_{s-1})$ for each step $s$, the standard Adam update rule applied to the parameters $\tilde{w}$ in this rotated space yields:

$$\tilde{w}_t = \tilde{w}_{t-1} - \eta_t \frac{\tilde{m}_t}{\sqrt{\tilde{v}_t + \epsilon}},$$

where the first and second moment estimates ($\tilde{m}_t$ and $\tilde{v}_t$) are computed using the gradients in the rotated space:

$$\tilde{m}_t = (1-\beta_1) \sum_{s=1}^{t} \beta_1^{t-s} \tilde{g}_s = (1-\beta_1) \sum_{s=1}^{t} \beta_1^{t-s} \mathcal{U}^\top \nabla f(w_{s-1}),$$
$$\tilde{v}_t = (1-\beta_2) \sum_{s=1}^{t} \beta_2^{t-s} \tilde{g}_s^2 = (1-\beta_2) \sum_{s=1}^{t} \beta_2^{t-s} (\mathcal{U}^\top \nabla f(w_{s-1}))^2.$$

To express this update trajectory in the original parameter space, we multiply both sides of the update equation by the orthogonal rotation matrix $\mathcal{U}$. Using the identity $\mathcal{U}\tilde{w} = \mathcal{U}\mathcal{U}^\top w = w$, we obtain:

$$\mathcal{U}\tilde{w}_t = \mathcal{U}\tilde{w}_{t-1} - \eta_t \mathcal{U} \frac{\tilde{m}_t}{\sqrt{\tilde{v}_t} + \epsilon},$$

$$w_t = w_{t-1} - \eta_t \mathcal{U} \frac{(1-\beta_1)\sum_{s=1}^{t} \beta_1^{t-s} \mathcal{U}^\top \nabla f(w_{s-1})}{\sqrt{(1-\beta_2)\sum_{s=1}^{t} \beta_2^{t-s}(\mathcal{U}^\top \nabla f(w_{s-1}))^2} + \epsilon}.$$

This strictly recovers Equation (6), confirming that Adam in the rotated space is equivalent to applying the rotation matrix $\mathcal{U}$ to the gradients, computing the coordinate-wise adaptive steps, and projecting the update back to the original space.

## D. Experimental Details

### D.1. Section 2

**Section 2.1: Figure 3**   To compare Adam's coordinate-wise adaptivity with SGD, we employed AdaSGD (Wang & Wiens, 2020), which scales a uniform learning rate across all parameters by taking the exponential moving average of the average second momentum. This setup enabled us to use an identical learning rate for both AdaSGD and Adam, facilitating a direct comparison of how gradient delay interacts with coordinate-wise adaptivity. We set the learning rate to $1.0$ and $\beta_1 = 0$ in order to isolate the adversarial update directions induced by delayed gradients without the confounding effect of momentum. For $\beta_2$, we used $0.1$ which is smaller than the standard Adam setting, to more clearly visualize oscillatory behavior. Nevertheless, our empirical analysis in Section 2.2 confirms that this alignment-dependent instability persists even with standard hyperparameters in practical training scenarios. Finally, we defined the convergence criterion as reaching a loss of $15.0$ for each optimizer, tested with and without a delay of $\tau = 2$.

**Section 2.2: Figure 4**   The spiral loss landscape was defined as $f(r, \theta) = r^2 + (20\sin(4r - \theta) + 1)^2$, where $r$ and $\theta$ represent the radius and angle in polar coordinates. We set the learning rate to $0.1$ and $\beta_1 = 0$ to focus on the effect of delayed gradients without momentum. We used $\beta_2 = 0.9$, matching standard Adam settings. Even with such a large $\beta_2$, Adam exhibited low oscillation when the Hessian eigenbasis is aligned with the standard coordinate basis, whereas severe oscillations emerge in the misaligned region. For Figure 4b, we introduced a delay of $\tau = 1$ at a randomly chosen iteration during training without delay. We then measured the number of iterations required to traverse an angular displacement of $3°$ with and without delay, and report their ratio as the slowdown metric.

### D.2. Section 4

Our model is based on nanoGPT (Karpathy, 2022). The model has an embedding dimension of $384$, sequence length of $512$, 6 attention heads, and 32 Transformer blocks. The model has learnable positional embedding and untied language model head. When the number of stages is $P$, we allocate $32/P$ Transformer blocks to each stage. The first and the last stage also hold embedding and language model head respectively. This model has approximately 96 million parameters.

The model is trained in bf16 mixed precision. We use the batch size of $8$, which is the maximum without incurring out-of-memory error in our setting, and do not use gradient accumulation since it changes the level of delay (Ajanthan et al., 2025). For each method and stage, we search the learning rate among $\{0.0001, 0.0003, 0.001\}$. $\beta_1$ is set to $0.99$ for Nesterov following the original paper (Ajanthan et al., 2025) and set to $0.9$ for others. The learning rate is scheduled using a linear warmup for the first $1.2\%$ of iterations, followed by a cosine decay for the remainder of the training process. We use $\beta_2$ of $0.999$, gradient clipping value of $1.0$, and weight decay of $0.01$. Unless specified otherwise, we set the basis update frequency to 10 for basis rotation. We only perform rotation to the MLP and attention layers excluding embedding layers, language model head, bias parameters, and layer normalization parameters. We use at most eight RTX 3090 GPUs for the experiments. When the number of pipeline stages $P$ exceeds the number of available physical GPUs, we configure multiple virtual stages to share a single GPU; for example, a 32-stage experiment on eight RTX 3090 GPUs assigns four virtual stages per GPU. This configuration preserves stage depth and gradient delay exactly, since the pipeline scheduling and communication order between forward, backward, and update operations are unchanged. While absolute hardware utilization and per-iteration wall-clock time may be affected by context-switching overhead, this overhead applies uniformly across all baselines and basis rotation, so the comparative results—including the loss vs. GPU hour comparison in Figure 9a—remain valid.

Our 1B model has an embedding dimension of 1728, sequence length of 512, 27 attention heads, and 24 Transformer blocks. The model has approximately 1.03 billion parameters. We use the learning rate of 0.0001. We use at most six A100 GPUs for these experiments.

### D.3. Empirical Validation in Figure 11

To observe the optimization trajectory and measure oscillation along specific eigenvector directions in Figure 11, we estimate the dominant eigenvector of the Hessian midway through the training process using power iteration. For the non-dominant direction, we sample a random vector and explicitly orthogonalize it against the estimated dominant eigenvector. We then track the projection of the parameter updates along these two isolated directions for a span of 100 iterations. For the Hessian $(1, 1)$-norm estimation, we follow the methodology of Xie et al. (2025), utilizing Hessian-vector products with random Cauchy vectors to approximate the norm without explicitly constructing the full Hessian matrix. Specifically, this estimation is rigorously computed over $8,000$ batches using $200$ sampled random Cauchy vectors. The resulting values are then normalized by dividing them by the total number of parameters in the model to provide a standardized metric for comparison.

## E. Proof of Theorem 2.3

In this section, we present the proof of Theorem 2.3. In the following, we use $H_i$ instead of $C_i$ defined in Assumption 2.2.

We first present several lemmas necessary for the proof. The lemmas are taken from Xie et al. (2025) for the case of Adam.

**Lemma E.1.** *(Lemma D.3. of Xie et al. (2025)) For any twice differentiable loss which is $H$-coordinate-wisely-smooth w.r.t. $\ell_\infty$ norm, we have for any $x$ and $\Delta \in \mathbb{R}^d$,*

$$|\Delta^\top \nabla^2 f(x)\Delta| \leq \sum_{i=1}^d H_i \Delta_i^2.$$

**Lemma E.2.** *(Lemma 3.12. of Xie et al. (2025)) For any $0 < \beta_2 < 1$, for any scalar sequences $\{v_t\}_{t=0}^T$ and $\{g_t\}_{t=1}^T$ satisfying $v_0 \geq 0, v_1 > 0$, and $v_t - \beta_2 v_{t-1} \geq (1-\beta_2)g_t^2$ for $t \geq 1$, the following holds:*

$$\sum_{t=1}^T \frac{g_t^2}{v_t} \leq T + \frac{\beta_2}{1-\beta_2} \ln \frac{v_T}{v_0}.$$

**Lemma E.3.** *(Lemma 3.13. of Xie et al. (2025)) Under Assumption 2.1, for any $i \in [d]$, the following holds:*

$$\mathbb{E}\sum_{t=1}^T \frac{g_{t,i}\bar{g}_{t,i}}{\sqrt{v_{t,i}+\epsilon}} \geq \frac{1}{2}\mathbb{E}\sum_{t=1}^T \frac{\bar{g}_{t,i}^2}{\sqrt{\tilde{v}_{t,i}+\epsilon}} - \sqrt{1-\beta_2}T\sigma_i - \frac{\sigma_i\beta_2}{\sqrt{1-\beta_2}}\mathbb{E}\left[\ln\frac{v_{T,i}+\epsilon}{v_{0,i}+\epsilon}\right],$$

*where $\tilde{v}$ is defined as $\tilde{v}_{t,i} = (1-\beta_2)(\bar{g}_{t,i}^2 + \sigma_i^2) + \beta_2 v_{t-1,i}$ and $g_t$ and $\bar{g}_t$ are stochastic the full-batch gradient respectively.*

**Lemma E.4.** *(Lemma 3.14. of Xie et al. (2025)) Under Assumption 2.1, for any $i \in [d]$, the following holds:*

$$\sum_{t=\frac{T}{2}+1}^T \mathbb{E}\left[\sqrt{\tilde{v}_{t,i}+\epsilon}\right] \leq \frac{2\beta_2^{\frac{T}{4}}}{1-\beta_2}\sqrt{v_{0,i}} + \frac{T}{2}\sigma_i + \frac{T}{2}\sqrt{\epsilon} + 2\sum_{t=1}^T \mathbb{E}\left[\frac{\bar{g}_{t,i}^2}{\sqrt{\tilde{v}_{t,i}+\epsilon}}\right]$$

**Lemma E.5.** *(Lemma D.1. of Xie et al. (2025)) With Assumptions 2.1 and 2.2, the following holds for any $T$:*

$$\ln\frac{\mathbb{E}\max_{i\in[d]} v_{T,i}+\epsilon}{v_0+\epsilon} \leq 2\ln\left(1 + \frac{\sum_{i=1}^d \sigma_i^2 + \|\nabla f(x_0)\|_\infty^2 + \sum_{i=1}^d H_i^2\eta^2 T(T+\frac{1}{1-\beta_2})}{v_0+\epsilon} + \ln 32\right)$$

Now, we are ready to prove Theorem 2.3. The key step for analyzing the convergence rate is to address the (B) term in Equation (1).

*Proof.* First, by the Taylor approximation, we have

$$f(x_t) = f(x_{t-1}) + \nabla f(x_{t-1})(x_t - x_{t-1}) + \frac{1}{2}(x_t - x_{t-1})^\top \nabla^2 f(x)(x_t - x_{t-1})$$

for some $x$. By Lemma E.1,

$$
\begin{aligned}
f(x_t) - f(x_{t-1}) &= \nabla f(x_{t-1})(x_t - x_{t-1}) + \frac{1}{2}\sum_{i=1}^{d} H_i(x_{t,i} - x_{t-1,i})^2 \\
&= -\eta \sum_{i=1}^{d} \frac{g_{t-\tau,i}\bar{g}_{t,i}}{\sqrt{v_{t-\tau,i}+\epsilon}} + \frac{1}{2}\eta^2 \sum_{i=1}^{d} H_i \frac{g_{t-\tau,i}^2}{v_{t-\tau,i}+\epsilon} \\
&= \underbrace{-\eta \sum_{i=1}^{d} \frac{\bar{g}_{t-\tau,i}g_{t-\tau,i}}{\sqrt{v_{t-\tau,i}+\epsilon}}}_{(A)} + \underbrace{\eta \sum_{i=1}^{d} \frac{(\bar{g}_{t-\tau,i} - \bar{g}_{t,i})g_{t-\tau,i}}{\sqrt{v_{t-\tau,i}+\epsilon}}}_{(B)} + \underbrace{\frac{1}{2}\eta^2 \sum_{i=1}^{d} H_i \frac{g_{t-\tau,i}^2}{v_{t-\tau,i}+\epsilon}}_{(C)} .
\end{aligned}
\tag{1}
$$

Note that we have added and subtracted $-\eta \sum_{i=1}^{d} \frac{\bar{g}_{t-\tau,i}g_{t-\tau,i}}{\sqrt{v_{t-\tau,i}+\epsilon}}$ so that we can use Lemma E.3. Since (A) and (C) can be upper bounded with Lemma E.3 and Lemma E.2 respectively, the remaining step is to upper bound (B).

By Cauchy-Schwarz inequality, we have

$$\eta \sum_{i=1}^{d} \frac{(\bar{g}_{t-\tau,i} - \bar{g}_{t,i})g_{t-\tau,i}}{\sqrt{v_{t-\tau,i}+\epsilon}} \leq \eta \underbrace{\sqrt{\sum_{i=1}^{d}(\bar{g}_{t-\tau,i} - \bar{g}_{t,i})^2}}_{A_t} \cdot \underbrace{\sqrt{\sum_{i=1}^{d} \frac{g_{t-\tau,i}^2}{v_{t-\tau,i}+\epsilon}}}_{B_t} .$$

Summing over $t$ and applying Cauchy-Schwarz inequality leads to

$$\eta \sum_{t=1}^{T+\tau} A_t B_t \leq \eta \sqrt{\sum_{t=1}^{T+\tau} A_t^2} \sqrt{\sum_{t=1}^{T+\tau} B_t^2} .$$

We can upper bound $\sqrt{\sum_{t=1}^{T+\tau} A_t^2}$ as follows.

$$
\begin{aligned}
\sqrt{\sum_{t=1}^{T+\tau} A_t^2} &= \sqrt{\sum_{t=1}^{T+\tau} \sum_{i=1}^{d} (\bar{g}_{t,i} - \bar{g}_{t-\tau,i})^2} \\
&\leq \sqrt{\sum_{t=1}^{T+\tau} \sum_{i=1}^{d} H_i^2 \|x_{t-1} - x_{t-\tau-1}\|_\infty^2} \\
&\leq \sqrt{\sum_{t=1}^{T+\tau} \sum_{i=1}^{d} H_i^2 \|x_{t-1} - x_{t-\tau-1}\|_2^2} \\
&= \sqrt{\sum_{t=1}^{T+\tau} \sum_{i=1}^{d} H_i^2 \left\| \sum_{k=1}^{\tau} \eta \frac{g_{t-k-\tau}}{\sqrt{v_{t-k-\tau} + \epsilon}} \right\|_2^2} \\
&\leq \sqrt{\sum_{t=1}^{T+\tau} \sum_{i=1}^{d} H_i^2 \tau \eta^2 \sum_{k=1}^{\tau} \left\| \frac{g_{t-k-\tau}}{\sqrt{v_{t-k-\tau} + \epsilon}} \right\|_2^2} \\
&= \sqrt{\tau \eta^2 \sum_{i=1}^{d} H_i^2 \sum_{t=1}^{T+\tau} \sum_{k=1}^{\tau} \left\| \frac{g_{t-k-\tau}}{\sqrt{v_{t-k-\tau} + \epsilon}} \right\|_2^2} \\
&\leq \sqrt{\tau \eta^2 \sum_{i=1}^{d} H_i^2 \tau \sum_{t=1}^{T+\tau} \left\| \frac{g_{t-\tau}}{\sqrt{v_{t-\tau} + \epsilon}} \right\|_2^2} \\
&= \tau \eta \sqrt{\sum_{i=1}^{d} H_i^2} \cdot \sqrt{\sum_{t=1}^{T+\tau} \left\| \frac{g_{t-\tau}}{\sqrt{v_{t-\tau} + \epsilon}} \right\|_2^2} \\
&= \tau \eta \sqrt{\sum_{i=1}^{d} H_i^2} \cdot \sqrt{\sum_{t=1}^{T} \sum_{i=1}^{d} \frac{g_{t,i}^2}{v_{t,i} + \epsilon}}
\end{aligned}
$$

We have used Assumption 2.2 for the first inequality. The second inequality holds because $\ell_\infty$ norm of a vector is always smaller than or equal to its $\ell_2$ norm. We have used Jensen's inequality for the third inequality. The final inequality holds because each term $\frac{g_{t-\tau}}{\sqrt{v_{t-\tau}+\epsilon}}$ is summed at most $\tau$ times.

Thus, we have

$$
\begin{aligned}
\eta \sum_{t=1}^{T} A_t B_t &\leq \eta \sqrt{\sum_{t=1}^{T} A_t^2} \sqrt{\sum_{t=1}^{T} B_t^2} \\
&\leq \tau \eta^2 \sqrt{\sum_{i=1}^{d} H_i^2} \cdot \sum_{t=1}^{T} \sum_{i=1}^{d} \frac{g_{t-\tau,i}^2}{v_{t-\tau,i} + \epsilon}.
\end{aligned}
\tag{2}
$$

Next, applying Lemma E.3 to (A) of Equation (1) gives

$$
\begin{aligned}
\sum_{t=1}^{T+\tau} \mathbb{E}(A) &= -\eta \sum_{t=1}^{T+\tau} \sum_{i=1}^{d} \mathbb{E} \frac{\bar{g}_{t-\tau,i} g_{t-\tau,i}}{\sqrt{v_{t-\tau,i}+\epsilon}} \\
&= -\eta \sum_{i=1}^{d} \sum_{t=1}^{T} \mathbb{E} \frac{\bar{g}_{t,i} g_{t,i}}{\sqrt{v_{t,i}+\epsilon}} \\
&\leq -\eta \sum_{i=1}^{d} \frac{1}{2} \mathbb{E} \sum_{t=1}^{T} \frac{\bar{g}_{t,i}^2}{\sqrt{\tilde{v}_{t,i}^2+\epsilon}} + \eta \sum_{i=1}^{d} \sqrt{1-\beta_2} T \sigma_i + \eta \sum_{i=1}^{d} \frac{\sigma_i \beta_2}{\sqrt{1-\beta_2}} \mathbb{E}\left[\ln \frac{v_{T,i}+\epsilon}{v_{0,i}+\epsilon}\right].
\end{aligned}
$$

Then, by summing over $t$ from $t = 0$ to $T_\tau$, dividing by $T$, applying Lemma E.2 to (C) and Equation (2), and rearranging the terms, we get the following.

$$
\begin{aligned}
l &= \frac{1}{T} \mathbb{E}\left[\sum_{i=1}^{d} \sum_{t=1}^{T} \frac{\bar{g}_{t,i}^2}{\sqrt{\tilde{v}_{t,i}+\epsilon}}\right] \\
&\leq \frac{2}{\eta T} \mathbb{E}[f(x_0) - f(x_{T+\tau})] + \frac{\eta}{T} \mathbb{E}\left[\sum_{i=1}^{d} H_i \left(T + \frac{\beta_2}{1-\beta_2} \ln \frac{v_{T,i}+\epsilon}{v_{0,i}+\epsilon}\right)\right] \\
&\quad + \frac{2}{T} \sum_{i=1}^{d} \sigma_i \sqrt{1-\beta_2} \left(T + \frac{\beta_2}{1-\beta_2} \mathbb{E} \ln \frac{v_{T,i}+\epsilon}{v_{0,i}+\epsilon}\right) \\
&\quad + \frac{2}{T} \tau \eta \sqrt{\sum_{i=1}^{d} H_i^2} \mathbb{E} \sum_{i=1}^{d} \left(T + \frac{\beta_2}{1-\beta_2} \ln \frac{v_{T,i}+\epsilon}{v_{0,i}+\epsilon}\right) \\
&\leq \frac{2}{\eta T} \mathbb{E}[f(x_0) - \min_x f(x)] + \eta \sum_{i=1}^{d} H_i + 2\sqrt{1-\beta_2} \sum_{i=1}^{d} \sigma_i + 2\tau \eta d \sqrt{\sum_{i=1}^{d} H_i^2} \\
&\quad + \frac{\beta_2}{T(1-\beta_2)} \left(\eta \sum_{i=1}^{d} H_i + 2\sqrt{1-\beta_2} \sum_{i=1}^{d} \sigma_i + 2\tau \eta d \sqrt{\sum_{i=1}^{d} H_i^2}\right) \max_i \mathbb{E} \ln \frac{v_{T,i}+\epsilon}{v_{0,i}+\epsilon} \\
&\leq \frac{2}{\eta T} \mathbb{E}[f(x_0) - \min_x f(x)] + \eta \sum_{i=1}^{d} H_i + 2\sqrt{1-\beta_2} \sum_{i=1}^{d} \sigma_i + 2\tau \eta d \sqrt{\sum_{i=1}^{d} H_i^2} \\
&\quad + \frac{\beta_2}{T(1-\beta_2)} \left(\eta \sum_{i=1}^{d} H_i + 2\sqrt{1-\beta_2} \sum_{i=1}^{d} \sigma_i + 2\tau \eta d \sqrt{\sum_{i=1}^{d} H_i^2}\right) \ln \frac{\mathbb{E} \max_i v_{T,i}+\epsilon}{v_0+\epsilon} \\
&\leq \frac{2}{\eta T} \mathbb{E}[f(x_0) - \min_x f(x)] + \eta(1+2\tau d) \sum_{i=1}^{d} H_i + 2\sqrt{1-\beta_2} \sum_{i=1}^{d} \sigma_i \\
&\quad + \leq \frac{\beta_2}{T(1-\beta_2)} \left(\eta(1+2\tau d) \sum_{i=1}^{d} H_i + 2\sqrt{1-\beta_2} \sum_{i=1}^{d} \sigma_i\right) \ln \frac{\mathbb{E} \max_i v_{T,i}+\epsilon}{v_0+\epsilon}.
\end{aligned}
$$

We have used $\sqrt{\sum_{i=1}^{d} H_i^2} \leq \sum_{i=1}^{d} H_i$ for the final inequality.

From Lemma E.5, we define

$$E = \frac{2}{\eta T} \mathbb{E}[f(x_0) - \min_x f(x)] + \eta(1 + 2\tau d) \sum_{i=1}^{d} H_i + 2\sqrt{1 - \beta_2} \sum_{i=1}^{d} \sigma_i$$
$$+ \frac{\beta_2}{T(1 - \beta_2)} \left( \eta(1 + 2\tau d) \sum_{i=1}^{d} H_i + 2\sqrt{1 - \beta_2} \sum_{i=1}^{d} \sigma_i \right) F$$

where

$$F = 2 \ln \left( 1 + \frac{\sum_{i=1}^{d} \sigma_i^2 + \|\nabla f(x_0)\|_\infty^2 + \sum_{i=1}^{d} H_i^2 \eta^2 T \left( T + \frac{1}{1 - \beta_2} \right)}{v_0 + \epsilon} \right) + \ln 32.$$

Then,

$$\frac{1}{T} \mathbb{E} \left[ \sum_{i=1}^{d} \sum_{t=1}^{T} \frac{\bar{g}_{t,i}^2}{\sqrt{\tilde{v}_{t,i} + \epsilon}} \right] \le E.$$

By Lemma E.4 and Cauchy-Schwarz inequality,

$$\frac{2}{T} \mathbb{E} \sum_{t=\frac{T}{2}+1}^{T} \sum_{i=1}^{d} |\bar{g}_{t,i}| \le \left( \frac{2}{T} \sum_{t=\frac{T}{2}+1}^{T} \sum_{i=1}^{d} \frac{\bar{g}_{t,i}^2}{\sqrt{\tilde{v}_{t,i} + \epsilon}} \right)^{1/2} \left( \frac{2}{T} \mathbb{E} \sum_{t=\frac{T}{2}+1}^{T} \sum_{i=1}^{d} \sqrt{\tilde{v}_{t,i} + \epsilon} \right)^{1/2}$$
$$\le \sqrt{2E} \left( 4E + \frac{4\beta_2^{T/4}}{T(1 - \beta_2)} d\sqrt{v_0} + \sum_{i=1}^{d} \sigma_i + d\sqrt{\epsilon} \right)^{1/2}$$
$$\le 2\sqrt{2}E + \sqrt{2}\sqrt{E} \sqrt{\frac{4\beta_2^{T/4}}{T(1 - \beta_2)} d\sqrt{v_0} + \sum_{i=1}^{d} \sigma_i + d\sqrt{\epsilon}}.$$

Finally, the following holds:

$$\min_{\frac{T}{2} < t \le T} \mathbb{E}[\|\nabla f(x_t)\|_1] \le \frac{1}{T} \sum_{t=\frac{T}{2}+1}^{T} \sum_{i=1}^{d} |\bar{g}_{t,i}|$$
$$\le \mathcal{O} \left( E + \sqrt{E} \sqrt{\frac{\beta_2^{T/4}}{T(1 - \beta_2)} d\sqrt{v_0} + \sum_{i=1}^{d} \sigma_i + d\sqrt{\epsilon}} \right)$$

with

$$E = \frac{2}{\eta T} \mathbb{E}[f(x_0) - f(x_{T+\tau})] + (1 + \frac{\beta_2 F}{T(1 - \beta_2)}) \left( \eta \sum_{i=1}^{d} H_i + 2\sqrt{1 - \beta_2} \sum_{i=1}^{d} \sigma_i + 2\tau\eta \sqrt{\sum_{i=1}^{d} H_i^2} \right)$$

and

$$F = 2 \ln \left( 1 + \frac{\sum_{i=1}^{d} \sigma_i^2 + \|\nabla f(x_0)\|_\infty^2 + \sum_{i=1}^{d} H_i^2 \eta^2 T \left( T + \frac{1}{1 - \beta_2} \right)}{v_0 + \epsilon} \right) + \ln 32.$$

When we assume $v_0 + \epsilon > \left( \sum_{i=1}^d \sigma_i^2 + \|\nabla f(x_0)\|_\infty^2 + \sum_{i=1}^d H_i^2 \eta^2 \right) / \text{poly}(T)$ and $\frac{1}{1-\beta_2} = \text{poly}(T)$, then $F = \mathcal{O}(\log T)$.

Terms involving $\sum_{i=1}^d \sigma_i$ have a lower bound $\Theta \left( \sum_{i=1}^d \sigma_i \left( \frac{\log T}{T} \right)^{1/2} \right)$ with $1 - \beta_2 = \Theta \left( \frac{\log T}{T} \right)$. Terms involving $\eta$ has

a lower bound $\Theta \left( \sqrt{\frac{(f(x_0) - \min_x f(x))(1+\tau) \sum_{i=1}^d H_i}{T}} \right)$ reached by $\eta = \Theta \left( \sqrt{\frac{f(x_0) - \min_x f(x)}{T(1+\tau d) \sum_{i=1}^d H_i}} \right)$.

For $R \triangleq (f(x_0) - \min_x f(x)) \sum_{i=1}^d H_i$, $\beta_1 = 0$, $1 - \beta_2 = \Theta \left( \frac{\log T}{T} \right)$, $\eta = \Theta \left( \sqrt{\frac{f(x_0) - \min_x f(x)}{T(1+\tau d) \sum_{i=1}^d H_i}} \right)$, $v_0 + \epsilon > \left( \sum_{i=1}^d \sigma_i^2 + \|\nabla f(x_0)\|_\infty^2 + \sum_{i=1}^d H_i^2 \eta^2 \right) / \text{poly}(T)$, and $\frac{1}{1-\beta_2} = \text{poly}(T)$, we have

$$\min_{\frac{T}{2} < t \leq T} \mathbb{E}\|\nabla f(x_t)\|_1 = \mathcal{O} \left( \sqrt{\frac{(1 + d\tau)R}{T}} + \sqrt{\sum_{i=1}^d \sigma_i \left( \frac{(1+d\tau)R}{T} \right)^{1/4}} + \sum_{i=1}^d \sigma_i \left( \frac{\log T}{T} \right)^{1/4} + \delta_T \right)$$

with $\delta_T = \sqrt{\frac{dv_0}{T(1-\beta_2)}} \exp\left( -\frac{T(1-\beta_2)}{8} \right) \left[ \left( \frac{(1+d\tau)R}{T} \right)^{1/4} + \sqrt{\sum_{i=1}^d \sigma_i \left( \frac{\log T}{T} \right)^{1/4}} \right]$. $\qquad \square$

### E.1. Extension to Stage-Dependent Delay

In this section, we extend the convergence analysis of Theorem 2.3 to accommodate stage-dependent delays, which more accurately reflects the structural mechanics of asynchronous pipeline parallelism.

Let $S_1, \ldots, S_K$ denote the partition of parameter coordinates across $K$ pipeline stages. For a coordinate $i \in S_k$, the gradient delay is determined by its stage distance, i.e., $\tau_i = K - k$. The update rule for coordinate $i$ is thus given by:

$$x_{t,i} = x_{t-1,i} - \eta \frac{g_{t_i,i}}{\sqrt{v_{t_i,i} + \epsilon}}$$

where $t_i = t - \tau_i$.

**Theorem E.6.** *Under the same assumptions as Theorem 2.3, the convergence rate of asynchronous Adam with stage-dependent delays $\{\tau_i\}$ is as follows:*

$$\min_{\frac{T}{2} < t \leq T} \mathbb{E}\|\nabla f(w_t)\|_1 = \mathcal{O} \left( \sqrt{\frac{(1+d\tau')\Delta_0 C}{T}} + \sqrt{\sum_{i=1}^d \sigma_i \left( \frac{(1+d\tau')\Delta_0 C}{T} \right)^{1/4}} + \sum_{i=1}^d \sigma_i \left( \frac{\log T}{T} \right)^{1/4} \right),$$

*where the effective delay $\tau'$ is defined as:*

$$\tau' \triangleq \sqrt{\frac{\sum_{i=1}^d C_i^2 \tau_i^2}{\sum_{i=1}^d C_i^2}} = \sqrt{\frac{\sum_{k=1}^K (K-k)^2 \sum_{i \in \mathcal{S}_k} C_i^2}{\sum_{i=1}^d C_i^2}}. \tag{15}$$

*Proof.* We follow the structure of the constant delay proof in Appendix E. Equation (1) is modified by replacing the constant $\tau$ with the coordinate-specific $\tau_i$. The bounds for terms (A) and (C) follow the exact same steps using Lemma E.3 and Lemma E.2.

For term (B), we redefine $A_t$ and $B_t$ by replacing $t - \tau$ with $t_i$:

$$A_t = \sqrt{\sum_{i=1}^d (\bar{g}_{t_i,i} - \bar{g}_{t,i})^2}, \quad B_t = \sqrt{\sum_{i=1}^d \frac{g_{t_i,i}^2}{v_{t_i,i} + \epsilon}}.$$

Applying the Cauchy-Schwarz inequality over the extended time horizon up to $T + K$ yields:

$$\eta \sum_{t=1}^{T+K} A_t B_t \leq \eta \sqrt{\sum_{t=1}^{T+K} A_t^2} \sqrt{\sum_{t=1}^{T+K} B_t^2}.$$

Next, we upper bound $\sum_{t=1}^{T+K} A_t^2$. Using Assumption 2.2, the property that the $\ell_\infty$ norm is bounded by the $\ell_2$ norm, and Jensen's inequality, we have:

$$
\begin{aligned}
\sum_{t=1}^{T+K} A_t^2 &= \sum_{t=1}^{T+K} \sum_{i=1}^{d} (\bar{g}_{t,i} - \bar{g}_{t_i,i})^2 \\
&\leq \sum_{t=1}^{T+K} \sum_{i=1}^{d} H_i^2 \|x_{t-1} - x_{t_i-1}\|_2^2 \\
&= \sum_{t=1}^{T+K} \sum_{i=1}^{d} H_i^2 \left\| \sum_{k=1}^{\tau_i} \eta \frac{g_{t-k_j,j}}{\sqrt{v_{t-k_j,j}+\epsilon}} \right\|_2^2 \\
&\leq \sum_{t=1}^{T+K} \sum_{i=1}^{d} H_i^2 \tau_i \eta^2 \sum_{k=1}^{\tau_i} \left\| \frac{g_{t-k_j,j}}{\sqrt{v_{t-k_j,j}+\epsilon}} \right\|_2^2 \\
&\leq \eta^2 \sum_{i=1}^{d} H_i^2 \tau_i^2 \sum_{t=1}^{T+K} \sum_{j=1}^{d} \frac{g_{t_j,j}^2}{v_{t_j,j}+\epsilon}
\end{aligned}
$$

The final inequality holds because each term $\frac{g_{t_j,j}}{\sqrt{v_{t_j,j}+\epsilon}}$ within the sliding window of size $\tau_i$ is summed at most $\tau_i$ times.

Therefore, taking the square root, we obtain:

$$
\sqrt{\sum_{t=1}^{T+K} A_t^2} \leq \eta \sqrt{\sum_{i=1}^{d} H_i^2 \tau_i^2} \cdot \sqrt{\sum_{t=1}^{T+K} \sum_{j=1}^{d} \frac{g_{t_j,j}^2}{v_{t_j,j}+\epsilon}}
$$

and consequently:

$$
\eta \sum_{t=1}^{T+K} A_t B_t \leq \eta^2 \sqrt{\sum_{i=1}^{d} H_i^2 \tau_i^2} \cdot \sum_{t=1}^{T+K} \sum_{j=1}^{d} \frac{g_{t_j,j}^2}{v_{t_j,j}+\epsilon}.
$$

Comparing this to the constant delay case in Equation (2), the term $\tau \sqrt{\sum_{i=1}^{d} H_i^2}$ is seamlessly replaced by $\sqrt{\sum_{i=1}^{d} H_i^2 \tau_i^2}$. We can define the effective delay $\tau'$ as:

$$
\tau' \triangleq \sqrt{\frac{\sum_{i=1}^{d} H_i^2 \tau_i^2}{\sum_{i=1}^{d} H_i^2}} = \sqrt{\frac{\sum_{k=1}^{K} (K-k)^2 \sum_{i \in S_k} H_i^2}{\sum_{i=1}^{d} H_i^2}}.
$$

This formulation allows us to rewrite $\sqrt{\sum_{i=1}^{d} H_i^2 \tau_i^2} = \tau' \sqrt{\sum_{i=1}^{d} H_i^2}$.

Following the remaining algebraic rearrangements as in Appendix E, and preserving the dimensionality dependence $d$ (which inherently arises from bounding the maximum over $d$ dimensions of the logarithmic variance terms), we arrive at the corresponding convergence bound:

$$
\min_{\frac{T}{2} < t \leq T} \mathbb{E} \|\nabla f(x_t)\|_1 = \mathcal{O}\left( \sqrt{\frac{(1+d\tau')R}{T}} + \sqrt{\sum_{i=1}^{d} \sigma_i} \left(\frac{(1+d\tau')R}{T}\right)^{1/4} + \sum_{i=1}^{d} \sigma_i \left(\frac{\log T}{T}\right)^{1/4} + \delta_T \right)
$$

where $R \triangleq (f(x_0) - \min_x f(x)) \sum_{i=1}^{d} H_i$.

This result demonstrates that modeling stage-dependent delays actively tightens the constant factor ($\tau' \leq K$, the maximum delay) while preserving the fundamental multiplicative interaction between the delay and the basis misalignment (represented by the $H_i$ terms). More importantly, the explicit formulation of $\tau'$ provides a rigorous theoretical validation for stage-aware basis rotation: by actively dampening the misalignment magnitude ($H_i^2$) in earlier pipeline stages where the delay $(K-k)$ is most severe, the effective delay $\tau'$ is directly minimized, yielding an accelerated convergence bound.

$\square$

**E.2. Extension to $\beta_1 > 0$**

In this section, we extend Theorem 2.3 to the heavy-ball momentum setting ($\beta_1 > 0$). The asynchronous Adam update with momentum is

$$m_{t,i} = \beta_1 \, m_{t-1,i} + (1 - \beta_1) \, g_{t,i}, \tag{16}$$

$$v_{t,i} = \beta_2 \, v_{t-1,i} + (1 - \beta_2) \, g_{t,i}^2, \tag{17}$$

$$x_{t,i} = x_{t-1,i} - \eta \, \frac{m_{t-\tau,i}}{\sqrt{v_{t-\tau,i} + \epsilon}}. \tag{18}$$

Equivalently, $m_s = (1 - \beta_1) \sum_{k=0}^{s-1} \beta_1^k g_{s-k}$. We extend the auxiliary variance shorthand to incorporate a momentum index: for $k \in \{0, 1, \ldots, s-1\}$,

$$\tilde{v}_{s,k+1,i} \triangleq \beta_2^{k+1} \, v_{s-k-1,i} + (1 - \beta_2) \, \mathbb{E}_{s-k-1}\left[ \sum_{j=s-k}^{s} \beta_2^{s-j} \, g_{j,i}^2 \right], \tag{19}$$

where $\mathbb{E}_{s-k-1}[\cdot]$ denotes conditional expectation w.r.t. $f_1, \ldots, f_{s-k-1}$. By construction, $\tilde{v}_{s,k+1,i}$ is $\sigma(f_1, \ldots, f_{s-k-1})$-measurable and therefore independent of $g_{s-k,i}, \ldots, g_{s,i}$. It reduces to $\tilde{v}_{s,i}$ of Lemma E.3 when $k = 0$.

For the heavy-ball extension we additionally adopt the standard coordinate-wise almost-sure gradient bound used in Défossez et al. (2022).

**Assumption E.7.** (Coordinate-wise bounded gradient) There exists $G > 0$ such that $\|\nabla f(w; \xi)\|_\infty \leq G$ almost surely for all $w \in \mathbb{R}^d$, where $\xi$ denotes the stochasticity of data.

**Theorem E.8** (Convergence with momentum). *Under Assumptions 2.1, 2.2 and E.7, for asynchronous Adam with $0 \leq \beta_1 < \beta_2 < 1$ where $\beta_1$ is a fixed constant independent of $T$, uniform delay $\tau$, and hyperparameter choices $1 - \beta_2 = \Theta(\log T / T)$ and $\eta = \Theta\left(\sqrt{\Delta_0 / (T \, d \, (1 + \tau) \sum_i H_i)}\right)$ with $\Delta_0 \triangleq f(x_0) - \min_x f(x)$, the following holds for all $T \geq T_0 = 4/(1 - \beta_1)$,*

$$\min_{T/2 < t \leq T} \mathbb{E}\|\nabla f(x_t)\|_1 = \mathcal{O}\left( \sqrt{\frac{d\,(1+\tau)\,R}{T}} + \sqrt{\sum_i \sigma_i} \left( \frac{d\,(1+\tau)\,R}{T} \right)^{1/4} + \sqrt{G\,d \sum_i \sigma_i} \left( \frac{\log T}{T} \right)^{1/4} \right),$$

*where $R \triangleq \Delta_0 \sum_{i=1}^d H_i$.*

E.2.1. AUXILIARY LEMMAS

The proof requires three additional lemmas. The first is a momentum-aware counterpart of Lemma E.3.

**Lemma E.9** (Hybrid descent lemma with momentum). *Under Assumptions 2.1, 2.2 and E.7, for any $i \in [d]$ and $0 \leq \beta_1 < \beta_2 \leq 1$,*

$$\mathbb{E} \sum_{s=1}^T \frac{\bar{g}_{s,i} \, m_{s,i}}{\sqrt{v_{s,i} + \epsilon}} \geq \frac{1 - \beta_1}{4} \, \mathbb{E} \sum_{s=1}^T \sum_{k=0}^{s-1} \beta_1^k \, \frac{\bar{g}_{s-k,i}^2}{\sqrt{\tilde{v}_{s,k+1,i} + \epsilon}}$$

$$- \frac{\beta_1 \, \eta \, H_i}{(1 - \beta_1)^2} \, \mathbb{E} \sum_{w=1}^T \left\| \frac{m_w}{\sqrt{v_w + \epsilon}} \right\|_2^2$$

$$- \frac{\sqrt{1 - \beta_1} \left[ 4G\sqrt{1 - \beta_2} + H_i \, \eta \right]}{(1 - \beta_1/\beta_2)^{3/2}} \, \mathbb{E} \sum_{u=1}^T \frac{g_{u,i}^2}{v_{u,i} + \epsilon}.$$

*Proof.* Fix coordinate $i$ and drop it from the notation; write $a_s := \sqrt{v_s + \epsilon}$ and $b_s^{(k)} := \sqrt{\tilde{v}_{s,k+1} + \epsilon}$. Using $m_s = (1 - \beta_1) \sum_{k=0}^{s-1} \beta_1^k g_{s-k}$ and the elementary identity $\bar{g}_s g_{s-k} = \bar{g}_{s-k} g_{s-k} + (\bar{g}_s - \bar{g}_{s-k}) g_{s-k}$,

$$\frac{\bar{g}_s m_s}{a_s} = (1 - \beta_1) \sum_{k=0}^{s-1} \beta_1^k \underbrace{\frac{\bar{g}_{s-k} \, g_{s-k}}{a_s}}_{(\mathcal{A}_k)} + (1 - \beta_1) \sum_{k=0}^{s-1} \beta_1^k \underbrace{\frac{(\bar{g}_s - \bar{g}_{s-k}) \, g_{s-k}}{a_s}}_{(\mathcal{B}_k)}. \tag{20}$$

We lower-bound the conditional expectation of $\mathcal{A}_k$ in Step A and upper-bound $|\mathcal{B}_k|$ in Step B.

**Step A: Bounding $\mathcal{A}_k$.** Define the partial sums

$$\delta_s^2 := (1 - \beta_2) \sum_{j=s-k}^{s} \beta_2^{s-j} g_j^2, \qquad r_s^2 := (1 - \beta_2) \sum_{j=s-k}^{s} \beta_2^{s-j} \mathbb{E}_{s-k-1}[g_j^2], \tag{21}$$

so that $a_s^2 = \beta_2^{k+1} v_{s-k-1} + \delta_s^2 + \epsilon$ and $\left(b_s^{(k)}\right)^2 = \beta_2^{k+1} v_{s-k-1} + r_s^2 + \epsilon$, hence $a_s^2 - \left(b_s^{(k)}\right)^2 = \delta_s^2 - r_s^2$, $a_s \geq \delta_s$, and $b_s^{(k)} \geq r_s$. Note that $r_s, \beta_2^{k+1} v_{s-k-1}, \tilde{v}_{s,k+1}$, and $b_s^{(k)}$ are all $\sigma(f_1, \ldots, f_{s-k-1})$-measurable. By the tower property,

$$\mathbb{E}\left[\frac{\bar{g}_{s-k} \, g_{s-k}}{b_s^{(k)}}\right] = \mathbb{E}\left[\frac{\bar{g}_{s-k}^2}{b_s^{(k)}}\right], \quad \text{so} \quad \mathbb{E}\left[\frac{\bar{g}_{s-k} \, g_{s-k}}{a_s}\right] = \mathbb{E}\left[\frac{\bar{g}_{s-k}^2}{b_s^{(k)}}\right] + \mathbb{E}[C_{s,k}], \tag{22}$$

where $C_{s,k} \triangleq \bar{g}_{s-k} \, g_{s-k} \left(1/a_s - 1/b_s^{(k)}\right)$. Using $|x - y| = |x^2 - y^2|/(x + y)$ and $a_s + b_s^{(k)} \geq \max(a_s, b_s^{(k)})$,

$$\left|\frac{1}{a_s} - \frac{1}{b_s^{(k)}}\right| = \frac{|a_s^2 - (b_s^{(k)})^2|}{a_s \, b_s^{(k)} \, (a_s + b_s^{(k)})} \leq \frac{r_s^2}{a_s \, (b_s^{(k)})^2} + \frac{\delta_s^2}{a_s^2 \, b_s^{(k)}}, \tag{23}$$

where the final inequality uses $|\delta_s^2 - r_s^2| \leq \delta_s^2 + r_s^2$. Hence $|C_{s,k}| \leq \kappa_{s,k} + \rho_{s,k}$ with

$$\kappa_{s,k} = \frac{|\bar{g}_{s-k} \, g_{s-k}| \, r_s^2}{a_s \, (b_s^{(k)})^2}, \qquad \rho_{s,k} = \frac{|\bar{g}_{s-k} \, g_{s-k}| \, \delta_s^2}{a_s^2 \, b_s^{(k)}}.$$

To bound $\kappa_{s,k}$, apply $|xy| \leq \frac{\lambda}{2} x^2 + \frac{1}{2\lambda} y^2$ with $\lambda = \frac{1}{2}\sqrt{1 - \beta_1} \, b_s^{(k)}$, $x = |\bar{g}_{s-k}|/b_s^{(k)}$, and $y = |g_{s-k}| \, r_s^2/(a_s \, b_s^{(k)})$:

$$\kappa_{s,k} \leq \frac{\sqrt{1 - \beta_1} \, \bar{g}_{s-k}^2}{4 \, b_s^{(k)}} + \frac{1}{\sqrt{1 - \beta_1}} \cdot \frac{g_{s-k}^2 \, r_s^4}{a_s^2 \, (b_s^{(k)})^3}. \tag{24}$$

Take $\mathbb{E}_{s-k-1}[\cdot]$. Since $r_s, b_s^{(k)}, \bar{g}_{s-k}$ are $\sigma(f_1, \ldots, f_{s-k-1})$-measurable, and using $(b_s^{(k)})^2 \geq r_s^2$ to bound $r_s^4/(b_s^{(k)})^3 \leq r_s^2/b_s^{(k)}$:

$$\mathbb{E}_{s-k-1}[\kappa_{s,k}] \leq \frac{\sqrt{1 - \beta_1} \, \bar{g}_{s-k}^2}{4 \, b_s^{(k)}} + \frac{r_s^2}{\sqrt{1 - \beta_1} \, b_s^{(k)}} \mathbb{E}_{s-k-1}\left[\frac{g_{s-k}^2}{a_s^2}\right]. \tag{25}$$

To bound $\rho_{s,k}$, apply the same inequality with $\lambda = \frac{1}{2}\sqrt{1 - \beta_1} \, b_s^{(k)}/r_s^2$, $x = |\bar{g}_{s-k}| \, \delta_s/b_s^{(k)}$, and $y = |\delta_s g_{s-k}|/a_s^2$:

$$\rho_{s,k} \leq \frac{\sqrt{1 - \beta_1} \, \bar{g}_{s-k}^2 \, \delta_s^2}{4 \, r_s^2 \, b_s^{(k)}} + \frac{r_s^2 \, \delta_s^2 \, g_{s-k}^2}{\sqrt{1 - \beta_1} \, b_s^{(k)} \, a_s^4}.$$

Taking $\mathbb{E}_{s-k-1}[\cdot]$, using $\mathbb{E}_{s-k-1}[\delta_s^2] = r_s^2$ on the first summand and $\delta_s^2 \leq a_s^2$ on the second:

$$\mathbb{E}_{s-k-1}[\rho_{s,k}] \leq \frac{\sqrt{1 - \beta_1} \, \bar{g}_{s-k}^2}{4 \, b_s^{(k)}} + \frac{r_s^2}{\sqrt{1 - \beta_1} \, b_s^{(k)}} \mathbb{E}_{s-k-1}\left[\frac{g_{s-k}^2}{a_s^2}\right]. \tag{26}$$

Adding Equations (25) and (26) and using $r_s^2/b_s^{(k)} \leq r_s$ together with $r_s \leq G\sqrt{(k+1)(1 - \beta_2)}$ (from Assumption E.7 and the geometric sum $\sum_{j=0}^{k} \beta_2^j \leq k + 1$):

$$\mathbb{E}_{s-k-1}[|C_{s,k}|] \leq \frac{\sqrt{1 - \beta_1} \, \bar{g}_{s-k}^2}{2 \, b_s^{(k)}} + \frac{2 \, G \, \sqrt{(k+1)(1 - \beta_2)}}{\sqrt{1 - \beta_1}} \mathbb{E}_{s-k-1}\left[\frac{g_{s-k}^2}{a_s^2}\right] \tag{27}$$

$$\leq \frac{\sqrt{1 - \beta_1}}{2} \mathbb{E}\left[\frac{\bar{g}_{s-k}^2}{b_s^{(k)}}\right] + \frac{2 \, G \, \sqrt{(k+1)(1 - \beta_2)}}{\sqrt{1 - \beta_1} \, \beta_2^k} \mathbb{E}\left[\frac{g_{s-k}^2}{v_{s-k} + \epsilon}\right], \tag{28}$$

where we use $a_s^2 \geq \beta_2^k(v_{s-k} + \epsilon)$ for the second inequality. Finally, we have

$$(1 - \beta_1) \sum_{s,k} \beta_1^k \mathbb{E}\left[\frac{\bar{g}_{s-k} \, g_{s-k}}{a_s}\right] \geq (1 - \beta_1) \sum_{s,k} \beta_1^k \left(\mathbb{E}\left[\frac{\bar{g}_{s-k}^2}{b_s^{(k)}}\right] - \mathbb{E}[|C_{s,k}|]\right) \tag{29}$$

$$\geq (1 - \beta_1) \sum_{s,k} \beta_1^k \left(\left(1 - \frac{\sqrt{1-\beta_1}}{2}\right) \mathbb{E}\left[\frac{\bar{g}_{s-k}^2}{b_s^{(k)}}\right] - \frac{2\,G\,\sqrt{(k+1)(1-\beta_2)}}{\sqrt{1-\beta_1}\,\beta_2^k} \mathbb{E}[\frac{g_{s-k}^2}{v_{s-k}+\epsilon}]\right) \tag{30}$$

$$\geq \frac{1-\beta_1}{2} \sum_{s,k} \beta_1^k \mathbb{E}\left[\frac{\bar{g}_{s-k}^2}{b_s^{(k)}}\right] - \frac{4\,G\,\sqrt{(1-\beta_1)(1-\beta_2)}}{(1-\beta_1/\beta_2)^{3/2}} \mathbb{E}\sum_u \frac{g_u^2}{v_u+\epsilon}, \tag{31}$$

where we use $1 - \sqrt{1-\beta_1}/2 \geq 1/2$ and Défossez et al. (2022, Lemma A.3), $\sum_{k \geq 0}(\beta_1/\beta_2)^k \sqrt{k+1} \leq 2/(1-\beta_1/\beta_2)^{3/2}$ in the last inequality.

**Step B: Bounding $\mathcal{B}_k$.** Apply $|xy| \leq \frac{\lambda}{2}x^2 + \frac{1}{2\lambda}y^2$ with $\lambda = \sqrt{1-\beta_1}/(H_i \eta \sqrt{k+1})$, $x = |\bar{g}_s - \bar{g}_{s-k}|$, $y = |g_{s-k}|/a_s$:

$$\sum_{s=1}^{T}\sum_{k=0}^{s-1} \beta_1^k |\mathcal{B}_k| \leq \sum_{s=1}^{T}\sum_{k=0}^{s-1} \beta_1^k \frac{\sqrt{1-\beta_1}}{2H_i\eta\sqrt{k+1}}(\bar{g}_{s,i} - \bar{g}_{s-k,i})^2 + \sum_{s=1}^{T}\sum_{k=0}^{s-1}\beta_1^k \frac{H_i\eta\sqrt{k+1}}{2\sqrt{1-\beta_1}} \cdot \frac{g_{s-k,i}^2}{a_s^2}. \tag{32}$$

The second term can be bound by

$$\sum_{s=1}^{T}\sum_{k=0}^{s-1}\beta_1^k \frac{H_i\eta\sqrt{k+1}}{2\sqrt{1-\beta_1}} \cdot \frac{g_{s-k,i}^2}{a_s^2} \leq \sum_{s=1}^{T}\sum_{k=0}^{s-1}\beta_1^k \frac{H_i\,\eta\,\sqrt{k+1}}{2\sqrt{1-\beta_1}\,\beta_2^k} \cdot \frac{g_{s-k,i}^2}{v_{s-k,i}+\epsilon} \tag{33}$$

$$\leq \frac{H_i\,\eta}{2\sqrt{1-\beta_1}} \sum_{u=1}^{T} \frac{g_{u,i}^2}{v_{u,i}+\epsilon} \sum_{k\geq 0}\left(\frac{\beta_1}{\beta_2}\right)^k \sqrt{k+1} \tag{34}$$

$$\leq \frac{H_i\,\eta}{(1-\beta_1/\beta_2)^{3/2}\sqrt{1-\beta_1}} \sum_{u=1}^{T} \frac{g_{u,i}^2}{v_{u,i}+\epsilon}, \tag{35}$$

where we use $a_s^2 \geq \beta_2^k(v_{s-k,i} + \epsilon)$ in the first inequality and Défossez et al. (2022, Lemma A.3) in the last. Taking expectation,

$$\mathbb{E}\,(1-\beta_1)\sum_{s=1}^{T}\sum_{k=0}^{s-1}\beta_1^k \frac{H_i\eta\sqrt{k+1}}{2\sqrt{1-\beta_1}} \cdot \frac{g_{s-k,i}^2}{a_s^2} \leq \frac{H_i\,\eta\,\sqrt{1-\beta_1}}{(1-\beta_1/\beta_2)^{3/2}} \mathbb{E}\sum_{u=1}^{T} \frac{g_{u,i}^2}{v_{u,i}+\epsilon}. \tag{36}$$

By Assumption 2.2 and Cauchy Schwarz,

$$(\bar{g}_{s,i} - \bar{g}_{s-k,i})^2 \leq H_i^2 \|x_{s-1} - x_{s-k-1}\|_\infty^2 \leq H_i^2\eta^2\Big(\sum_{l=1}^{k}\|\tfrac{m_{s-l}}{\sqrt{v_{s-l}+\epsilon}}\|_\infty\Big)^2 \leq H_i^2\eta^2 k\sum_{l=1}^{k}\|\tfrac{m_{s-l}}{\sqrt{v_{s-l}+\epsilon}}\|_\infty^2. \tag{37}$$

Substituting Equation (37) into the first term of Equation (32) gives

$$(1-\beta_1)\sum_{s=1}^{T}\sum_{k=0}^{s-1}\beta_1^k \frac{\sqrt{1-\beta_1}}{2H_i\eta\sqrt{k+1}}(\bar{g}_{s,i} - \bar{g}_{s-k,i})^2 \leq \frac{H_i\,(1-\beta_1)^{3/2}\,\eta}{2}\sum_{s,k}\beta_1^k \frac{k}{\sqrt{k+1}}\sum_{l=1}^{k}\|\tfrac{m_{s-l}}{\sqrt{v_{s-l}+\epsilon}}\|_\infty^2 \tag{38}$$

$$\leq \frac{H_i\,(1-\beta_1)^{3/2}\,\eta}{2}\sum_{u=1}^{T}\|\tfrac{m_u}{\sqrt{v_u+\epsilon}}\|_\infty^2 \sum_{k\geq 1}\beta_1^k \frac{k^2}{\sqrt{k+1}} \tag{39}$$

$$\leq \frac{H_i\,(1-\beta_1)^{3/2}\,\eta}{2} \cdot \frac{2\,\beta_1}{(1-\beta_1)^{5/2}}\sum_{u=1}^{T}\|\tfrac{m_u}{\sqrt{v_u+\epsilon}}\|_\infty^2 \tag{40}$$

$$= \frac{\beta_1\,H_i\,\eta}{1-\beta_1}\sum_{u=1}^{T}\|\tfrac{m_u}{\sqrt{v_u+\epsilon}}\|_\infty^2, \tag{41}$$

where we apply $k^2/\sqrt{k+1} \leq k^{3/2}$ for $k \geq 1$ and Défossez et al. (2022, Lemma A.4) in the last inequality.

Finally, from the fact that $\| \cdot \|_\infty^2 \leq \| \cdot \|_2^2$, we have

$$(1-\beta_1) \sum_{s=1}^{T} \sum_{k=0}^{s-1} \beta_1^k \frac{\sqrt{1-\beta_1}}{2H_i\eta\sqrt{k+1}} (\bar{g}_{s,i} - \bar{g}_{s-k,i})^2 \leq \frac{\beta_1 H_i \eta}{1-\beta_1} \sum_{u=1}^{T} \| \tfrac{m_u}{\sqrt{v_u+\epsilon}} \|_2^2. \tag{42}$$

Combining Equations (31), (36) and (42) completes the proof. $\qquad\square$

The second lemma is a momentum-aware counterpart of Lemma E.2.

**Lemma E.10** (Momentum variance bound). *For any $i \in [d]$ and $0 \leq \beta_1 < \beta_2 \leq 1$,*

$$\sum_{s=1}^{T} \frac{m_{s,i}^2}{v_{s,i}+\epsilon} \leq \frac{1-\beta_1}{1-\beta_1/\beta_2} \sum_{s=1}^{T} \frac{g_{s,i}^2}{v_{s,i}+\epsilon} \leq \frac{1-\beta_1}{1-\beta_1/\beta_2} \left( T + \frac{\beta_2}{1-\beta_2} \ln \frac{v_{T,i}+\epsilon}{v_{0,i}+\epsilon} \right).$$

*Proof.* By Cauchy–Schwarz applied to $m_{s,i} = (1-\beta_1) \sum_{u=1}^{s} \beta_1^{s-u} g_{u,i}$,

$$m_{s,i}^2 \leq (1-\beta_1)^2 \left( \sum_{u=1}^{s} \beta_1^{s-u} \right) \left( \sum_{u=1}^{s} \beta_1^{s-u} g_{u,i}^2 \right) \leq (1-\beta_1) \sum_{u=1}^{s} \beta_1^{s-u} g_{u,i}^2.$$

For $u \leq s$, the recursion for $v$ gives $v_{s,i} \geq \beta_2^{s-u} v_{u,i}$, hence $v_{s,i} + \epsilon \geq \beta_2^{s-u}(v_{u,i}+\epsilon)$. Therefore

$$\sum_{s=1}^{T} \frac{m_{s,i}^2}{v_{s,i}+\epsilon} \leq (1-\beta_1) \sum_{u=1}^{T} \frac{g_{u,i}^2}{v_{u,i}+\epsilon} \sum_{s\geq u} \left( \tfrac{\beta_1}{\beta_2} \right)^{s-u} = \frac{1-\beta_1}{1-\beta_1/\beta_2} \sum_{u=1}^{T} \frac{g_{u,i}^2}{v_{u,i}+\epsilon}.$$

$\qquad\square$

The third lemma extends Lemma E.4 to handle the $\beta_1^k$-weighted sum over the momentum index.

**Lemma E.11** (Momentum-weighted analog of Lemma E.4). *Under Assumption 2.2, for any $i \in [d]$ and $0 \leq \beta_1 < \beta_2 < 1$,*

$$\sum_{s=T/2+1}^{T} \sum_{k=0}^{s-1} \beta_1^k \mathbb{E}\left[ \sqrt{\tilde{v}_{s,k+1,i}+\epsilon} \right] \leq \frac{1}{1-\beta_1} \left[ \frac{2\beta_2^{T/4}}{1-\beta_2} \sqrt{v_{0,i}} + \frac{T}{2}\sigma_i + \frac{T}{2}\sqrt{\epsilon} + 2\sum_{s=1}^{T} \mathbb{E}\left[ \frac{\bar{g}_{s,i}^2}{\sqrt{\tilde{v}_{s,1,i}+\epsilon}} \right] \right].$$

*Proof.* Fix $k \in \{0, \ldots, s-1\}$. By definition of $\tilde{v}_{s,k+1,i}$ in Equation (19), the proof of Lemma E.4 (Xie et al. (2025, Lemma 3.16)) applies verbatim to the inner sum $\sum_{s=T/2+1}^{T} \mathbb{E}[\sqrt{\tilde{v}_{s,k+1,i}+\epsilon}]$ at each fixed $k$, yielding

$$\sum_{s=T/2+1}^{T} \mathbb{E}\left[ \sqrt{\tilde{v}_{s,k+1,i}+\epsilon} \right] \leq \frac{2\beta_2^{T/4}}{1-\beta_2} \sqrt{v_{0,i}} + \frac{T}{2}\sigma_i + \frac{T}{2}\sqrt{\epsilon} + 2\sum_{s=1}^{T} \mathbb{E}\left[ \frac{\bar{g}_{s,i}^2}{\sqrt{\tilde{v}_{s,k+1,i}+\epsilon}} \right]. \tag{43}$$

Multiplying equation 43 by $\beta_1^k$ and summing over $k$,

$$\sum_{s=T/2+1}^{T} \sum_{k=0}^{s-1} \beta_1^k \mathbb{E}\left[ \sqrt{\tilde{v}_{s,k+1,i}+\epsilon} \right] \leq \frac{1}{1-\beta_1} \left[ \frac{2\beta_2^{T/4}}{1-\beta_2} \sqrt{v_{0,i}} + \frac{T}{2}\sigma_i + \frac{T}{2}\sqrt{\epsilon} \right]$$

$$+ 2\sum_{s=1}^{T} \sum_{k=0}^{s-1} \beta_1^k \mathbb{E}\left[ \frac{\bar{g}_{s,i}^2}{\sqrt{\tilde{v}_{s,k+1,i}+\epsilon}} \right].$$

Since $\tilde{v}_{s,k+1,i}$ is non-decreasing in $k$ for fixed $s$ (its definition replaces a longer prefix of stochastic gradients by their conditional means), $\sqrt{\tilde{v}_{s,k+1,i}+\epsilon} \geq \sqrt{\tilde{v}_{s,1,i}+\epsilon}$, which gives $1/\sqrt{\tilde{v}_{s,k+1,i}+\epsilon} \leq 1/\sqrt{\tilde{v}_{s,1,i}+\epsilon}$. Hence the last double sum is at most $\sum_{s} \mathbb{E}[\bar{g}_{s,i}^2/\sqrt{\tilde{v}_{s,1,i}+\epsilon}] \cdot \sum_{k\geq0} \beta_1^k = \frac{1}{1-\beta_1} \sum_{s} \mathbb{E}[\bar{g}_{s,i}^2/\sqrt{\tilde{v}_{s,1,i}+\epsilon}]$, giving the lemma. $\qquad\square$

E.2.2. PROOF OF THEOREM E.8

*Proof.* **(1)** Following the procedure of Equation (1),

$$\eta \, \mathbb{E} \sum_{t,i} \underbrace{\frac{\bar{g}_{t-\tau,i} \, m_{t-\tau,i}}{\sqrt{v_{t-\tau,i} + \epsilon}}}_{(A)_{t,i}} \;\leq\; \Delta_0 + \eta \Big| \mathbb{E} \sum_{t,i} \underbrace{\frac{(\bar{g}_{t-\tau,i} - \bar{g}_{t,i}) \, m_{t-\tau,i}}{\sqrt{v_{t-\tau,i} + \epsilon}}}_{(B)_{t,i}} \Big| + \frac{\eta^2}{2} \, \mathbb{E} \sum_{t,i} H_i \underbrace{\frac{m_{t-\tau,i}^2}{v_{t-\tau,i} + \epsilon}}_{(C)_{t,i}} . \tag{44}$$

Applying Lemma E.9 per coordinate and summing over $i$:

$$\eta \, \mathbb{E} \sum_{t,i} (A)_{t,i} \;\geq\; \tfrac{(1-\beta_1)\eta}{4} \, \mathcal{S} - \eta R_H - \eta R_g, \tag{45}$$

where $\mathcal{S} \triangleq \mathbb{E} \sum_{s=1}^{T} \sum_{i=1}^{d} \sum_{k=0}^{s-1} \beta_1^k \, \bar{g}_{s-k,i}^2 / \sqrt{\tilde{v}_{s,k+1,i} + \epsilon}$ and

$$R_H \triangleq \frac{\beta_1 \eta \sum_i H_i}{(1-\beta_1)^2} \, \mathbb{E} \sum_{w=1}^{T} \Big\| \tfrac{m_w}{\sqrt{v_w + \epsilon}} \Big\|_2^2, \quad R_g \triangleq \frac{\sqrt{1-\beta_1}}{(1 - \beta_1/\beta_2)^{3/2}} \sum_{i=1}^{d} \big[ 4G\sqrt{1-\beta_2} + H_i \eta \big] \, \mathbb{E} \sum_{u=1}^{T} \frac{g_{u,i}^2}{v_{u,i} + \epsilon} .$$

For $(B)$, by Cauchy–Schwarz applied coordinatewise, $| \sum_i (B)_{t,i} | \leq A_t B_t$ where

$$A_t \triangleq \sqrt{\sum_i (\bar{g}_{t-\tau,i} - \bar{g}_{t,i})^2}, \qquad B_t \triangleq \sqrt{\sum_i m_{t-\tau,i}^2 / (v_{t-\tau,i} + \epsilon)} .$$

By smoothness, $(\bar{g}_{t,i} - \bar{g}_{t-\tau,i})^2 \leq H_i^2 \|x_{t-1} - x_{t-\tau-1}\|_\infty^2 \leq H_i^2 \|x_{t-1} - x_{t-\tau-1}\|_2^2$, and since the algorithm step is now $x_t - x_{t-1} = -\eta \, m_{t-\tau} / \sqrt{v_{t-\tau} + \epsilon}$,

$$\|x_{t-1} - x_{t-\tau-1}\|_2^2 \;=\; \eta^2 \Big\| \sum_{k=1}^{\tau} \tfrac{m_{t-k-\tau}}{\sqrt{v_{t-k-\tau} + \epsilon}} \Big\|_2^2 \;\leq\; \tau \eta^2 \sum_{k=1}^{\tau} \Big\| \tfrac{m_{t-k-\tau}}{\sqrt{v_{t-k-\tau} + \epsilon}} \Big\|_2^2 .$$

Using $\sum_{t,k} \|m_{t-k-\tau} / \sqrt{v_{t-k-\tau} + \epsilon}\|_2^2 \leq \tau \sum_w \|m_w / \sqrt{v_w + \epsilon}\|_2^2$ yields $\sum_t A_t^2 \leq \tau^2 \eta^2 \sum_i H_i^2 \sum_w \|m_w / \sqrt{v_w + \epsilon}\|_2^2$, while $\sum_t B_t^2 = \sum_w \|m_w / \sqrt{v_w + \epsilon}\|_2^2$ after re-indexing. Cauchy–Schwarz over $t$:

$$\eta \Big| \mathbb{E} \sum_{t,i} (B)_{t,i} \Big| \;\leq\; \eta \, \mathbb{E} \sqrt{\sum_t A_t^2} \sqrt{\sum_t B_t^2} \;\leq\; \tau \eta^2 \sqrt{\sum_i H_i^2} \, \mathbb{E} \sum_w \Big\| \tfrac{m_w}{\sqrt{v_w + \epsilon}} \Big\|_2^2 .$$

Applying Lemma E.10 to convert $\|m_w / \sqrt{v_w + \epsilon}\|_2^2 \to \|g_w / \sqrt{v_w + \epsilon}\|_2^2$ at the cost of $(1-\beta_1)/(1 - \beta_1/\beta_2)$:

$$\eta \Big| \mathbb{E} \sum_{t,i} (B)_{t,i} \Big| \;\leq\; \frac{(1-\beta_1) \, \tau \eta^2 \sqrt{\sum_i H_i^2}}{1 - \beta_1/\beta_2} \, \mathbb{E} \sum_{t,i} \frac{g_{t,i}^2}{v_{t,i} + \epsilon} . \tag{46}$$

For $(C)$, re-indexing $w = t - \tau$ and applying Lemmas E.2 and E.10 per coordinate:

$$\frac{\eta^2}{2} \, \mathbb{E} \sum_{t,i} H_i \, (C)_{t,i} \leq \frac{(1-\beta_1)\eta^2}{2(1 - \beta_1/\beta_2)} \sum_i H_i \Big( T + \tfrac{\beta_2}{1-\beta_2} \, \mathbb{E} \ln \tfrac{v_{T,i} + \epsilon}{v_{0,i} + \epsilon} \Big) . \tag{47}$$

**(2)** By Lemma E.2, $\sum_w g_{w,i}^2 / (v_{w,i} + \epsilon) \leq T + \beta_2 F_i / (1 - \beta_2)$ for each $i$ with $F_i \triangleq \ln((v_{T,i} + \epsilon)/(v_{0,i} + \epsilon))$, and Lemma E.5 gives $\mathbb{E} \sum_i F_i \leq dF$ with $F = \mathcal{O}(\log T)$. Setting $K_T \triangleq 1 + \beta_2 F / (T(1 - \beta_2)) = \mathcal{O}(1)$,

$$\mathbb{E} \sum_{w,i} \frac{g_{w,i}^2}{v_{w,i} + \epsilon} \;\leq\; dT K_T, \qquad \mathbb{E} \sum_i H_i \sum_w \frac{g_{w,i}^2}{v_{w,i} + \epsilon} \;\leq\; T K_T \sum_i H_i . \tag{48}$$

Applying equation 48 to the right-hand sides of Equations (45) to (47):

$$\eta R_H \;\leq\; \frac{\beta_1 d\eta^2 T K_T \sum_i H_i}{(1-\beta_1)(1-\beta_1/\beta_2)}, \tag{49}$$

$$\eta R_g \;\leq\; \frac{4Gd\eta T K_T \sqrt{(1-\beta_1)(1-\beta_2)}}{(1-\beta_1/\beta_2)^{3/2}} + \frac{\eta^2\sqrt{1-\beta_1}\,T K_T \sum_i H_i}{(1-\beta_1/\beta_2)^{3/2}}, \tag{50}$$

$$\eta\Big|\mathbb{E}\sum_{t,i}(B)_{t,i}\Big| \;\leq\; \frac{(1-\beta_1)d\tau\eta^2 T K_T \sum_i H_i}{1-\beta_1/\beta_2}, \tag{51}$$

$$\frac{\eta^2}{2}\mathbb{E}\sum_{t,i} H_i(C)_{t,i} \;\leq\; \frac{(1-\beta_1)\eta^2 T K_T \sum_i H_i}{2(1-\beta_1/\beta_2)}. \tag{52}$$

Substituting Equations (49) to (52) into equation 44:

$$\frac{(1-\beta_1)\mathcal{S}}{T} \;\leq\; \mathcal{E}_{\beta_1} \;\triangleq\; \frac{4\Delta_0}{\eta T} + K_T\Big[\tfrac{c_3(1-\beta_1)\eta(1+d\tau)\sum_i H_i}{1-\beta_1/\beta_2} + \tfrac{c_4\,\beta_1 d\eta\sum_i H_i}{(1-\beta_1)(1-\beta_1/\beta_2)} + \tfrac{c_5 Gd\sqrt{(1-\beta_1)(1-\beta_2)}}{(1-\beta_1/\beta_2)^{3/2}}\Big], \tag{53}$$

where $c_3, c_4, c_5 > 0$ are absolute constants.

**(3)** Let $\mathcal{E}' \triangleq \frac{2}{T}\mathbb{E}\sum_{s>T/2}\sum_{i,k}\beta_1^k\sqrt{\tilde{v}_{s,k+1,i}+\epsilon}$ and $\bar{w} \triangleq \sum_{j=T/2+1}^{T}(1-\beta_1^{T-j+1})/(1-\beta_1)$. Re-indexing $j = s - k$ gives $\sum_{s>T/2,k}\beta_1^k|\bar{g}_{s-k,i}| \geq \sum_{j>T/2} w_j|\bar{g}_{j,i}|$ with $w_j = (1-\beta_1^{T-j+1})/(1-\beta_1)$. By min-vs-weighted-average and Cauchy–Schwarz over $(i,s,k)$ $(\beta_1^k = \beta_1^{k/2}\cdot\beta_1^{k/2})$:

$$\min_{T/2<t\leq T}\mathbb{E}\|\nabla f(x_t)\|_1 \leq \tfrac{1}{\bar{w}}\mathbb{E}\sum_i\sum_{s>T/2}\sum_k \beta_1^k|\bar{g}_{s-k,i}| \leq \tfrac{1}{\bar{w}}\sqrt{\mathcal{S}\cdot T\mathcal{E}'/2}$$

$$\leq 2\sqrt{2}\,\sqrt{(1-\beta_1)\mathcal{E}_{\beta_1}\,\mathcal{E}'}, \tag{54}$$

where the last step uses $\bar{w} = \frac{T}{2(1-\beta_1)} - \frac{\beta_1(1-\beta_1^{T/2})}{(1-\beta_1)^2} \geq \frac{T}{4(1-\beta_1)}$ (holds for $T \geq T_0 = 4/(1-\beta_1)$) and $(1-\beta_1)\mathcal{S}/T \leq \mathcal{E}_{\beta_1}$ from equation 53. By Lemma E.11,

$$\mathcal{E}' \;\leq\; \mathcal{O}\big(\tfrac{\sum_i \sigma_i}{1-\beta_1} + \tfrac{\mathcal{E}_{\beta_1}}{(1-\beta_1)^2} + \tfrac{1}{T(1-\beta_1)}\big). \tag{55}$$

Substituting equation 55 into equation 54 and using $\sqrt{a+b+c} \leq \sqrt{a} + \sqrt{b} + \sqrt{c}$:

$$\min_{T/2<t\leq T}\mathbb{E}\|\nabla f(x_t)\|_1 \;\lesssim\; \sqrt{\mathcal{E}_{\beta_1}\sum_i \sigma_i} + \frac{\mathcal{E}_{\beta_1}}{\sqrt{1-\beta_1}} + \sqrt{\mathcal{E}_{\beta_1}/T}. \tag{56}$$

Let $D_\beta \triangleq (1-\beta_1)(1+d\tau) + \beta_1 d/(1-\beta_1) = \Theta(d(1+\tau))$. With $\eta = \Theta(\sqrt{\Delta_0/(TD_\beta\sum_i H_i)})$ and $1-\beta_2 = \Theta(\log T/T)$, equation 53 balances to $\mathcal{E}_{\beta_1} = \mathcal{O}(\sqrt{d(1+\tau)R/T} + Gd\sqrt{\log T/T})$. Substituting into equation 56 yields the theorem rate. $\qquad\square$

# F. Proof of Basis Misalignment Analysis

In this section, we provide the proof of the Hessian approximation comparison in Section 3.2. Before we begin the proof, we present useful lemmas regarding the Kronecker product.

**Lemma F.1.** *(Henderson & Searle (1981)) Let $A, B, C$ be matrices of appropriate dimensions. Then the following holds:*

$$\mathrm{vec}(ABC) = (C^\top \otimes A)\,\mathrm{vec}(B)$$

**Lemma F.2.** *(Horn & Johnson (1991)) Let $A, B, A', B'$ be matrices of appropriate dimensions. Then the followings hold:*

1. $(A \otimes B)(A' \otimes B') = (AA') \otimes (BB')$
2. $(A \otimes B)^\top = (A^\top \otimes B^\top)$

**Lemma F.3.**

$$\|A \otimes B\|_{(1,1)} = \|A\|_{(1,1)}\|B\|_{(1,1)}$$

*Proof.* $\|A \otimes B\|_{(1,1)} = \sum_{i,j,k,l}|A_{ik}B_{jl}| = \big(\sum_{ik}|A_{ik}|\big)\big(\sum_{jl}|B_{jl}|\big) = \|A\|_{(1,1)}\|B\|_{(1,1)}.$ $\qquad\square$

### F.1. Proof of Theorem 3.1

Let us restate the theorem here for the sake of readability.

**Theorem 3.1.** *Let $U$ and $V$ be the matrices whose columns are eigenvectors of $\mathbb{E}[GG^\top]$ and $\mathbb{E}[G^\top G]$ respectively. If Hessian admits a Kronecker-factorized empirical Fisher, the following inequalities hold:*

$$||H_{U,V}||_{(1,1)} \leq ||H_U||_{(1,1)} \leq ||H||_{(1,1)}.$$

*Moreover, $\|H_{U,V}\|_{(1,1)}$ achieves the global minimum over all rotations.*

In this proof, we rely on the key property of the Hessian with exact Kronecker product form.

**Lemma F.4.** *if $\mathbb{E}[gg^\top] = A \otimes B$ for some $A \in \mathbb{R}^{n \times n}$ and $B \in \mathbb{R}^{m \times m}$, then*

$$\mathbb{E}[gg^\top] = c \cdot \mathbb{E}[G^\top G] \otimes \mathbb{E}[GG^\top]$$

*for some scalar c.*

*Proof.* By the definition of Kronecker product, $\forall i, j, k, l : \mathbb{E}[G_{ij}G_{kl}] = A_{jl}B_{ik}$ holds. Then each entry of the matrices $\mathbb{E}[GG^\top]$ and $\mathbb{E}[G^\top G]$ satisfies

$$(\mathbb{E}[GG^\top])_{ik} = \sum_j \mathbb{E}[G_{ij}G_{kj}] = \sum_j A_{jj}B_{ik} = \text{Tr}(A) \cdot B_{ik}$$

$$(\mathbb{E}[G^\top G])_{jl} = \sum_i \mathbb{E}[G_{ij}G_{il}] = \sum_i A_{jl}B_{ii} = A_{jl} \cdot \text{Tr}(B).$$

Thus, $\mathbb{E}[GG^\top]$ and $\mathbb{E}[G^\top G]$ are scalar multiplications of $B$ and $A$, respectively. $\qquad\square$

Now, we are ready to prove Theorem 3.1.

*Proof.* Since Hessian admits a Kroneker factorized empirical Fisher, $H = \mathbb{E}[gg^\top] = A \otimes B$ holds with some $A \in \mathbb{R}^{n \times n}$ and $B \in \mathbb{R}^{m \times m}$. From Lemma F.4, and the definitions of $U$ and $V$ in the theorem, we can write the eigendecompositions as:

$$A = V\Lambda_A V^\top, \quad B = U\Lambda_B U^\top,$$

where $\Lambda_A$ and $\Lambda_B$ are diagonal matrices.

By Lemma F.1, rotated parameterizations corresponds to $\tilde{w} = (I \otimes U^\top)w$ and $\tilde{w} = (V^\top \otimes U^\top)w$.

Then Hessian in the rotated space is

$$H_U = (I \otimes U^\top)(A \otimes B)(I \otimes U) = A \otimes (U^\top BU) = A \otimes \Lambda_B, \tag{57}$$

$$H_{U,V} = (V^\top \otimes U^\top)(A \otimes B)(V \otimes U) = (V^\top AV) \otimes (U^\top BU) = \Lambda_A \otimes \Lambda_B, \tag{58}$$

where in the second equalities, we use Lemma F.2 and in the last equalities, we used Lemma F.4 in both (58) and (57). For any symmetric matrix, the $(1,1)$-norm is minimized when the matrix is diagonalized by its eigenbasis. Therefore:

$$||\Lambda_A||_{(1,1)} \leq ||A||_{(1,1)} \quad \text{and} \quad ||\Lambda_B||_{(1,1)} \leq ||B||_{(1,1)}.$$

Thus, Lemma F.3 concludes the theorem. $\qquad\square$

### F.2. Basis misalignment Analysis for $\mathcal{S} = 1^{\text{st}}$

Here, we present a result for the $\mathcal{S} = 1^{\text{st}}$ strategy analogous to Theorem 3.1, but under a different assumption regarding the Hessian's structure. Under the assumption $H = \mathbb{E}[g]\mathbb{E}[g]^\top$, we can show the following.

**Theorem F.5.** *Let orthogonal matrices $U, V$ be $\mathbb{E}[G] = U\Sigma V^\top$. If Hessian has an exact Kronecker product form, the following inequalities hold:*

$$||H_{U,V}||_{(1,1)} \le ||H_U||_{(1,1)} \le ||H||_{(1,1)}.$$

*Moreover, $\|H_{U,V}\|_{(1,1)}$ attains the global minimum over all orthogonal rotations.*

However, in practice, $\mathbb{E}[g]\mathbb{E}[g]^\top$ is not guaranteed to be a faithful approximation of the true Hessian. This discrepancy potentially limits the approximation fidelity and, consequently, the optimizer's robustness to gradient delay in empirical settings.

*Proof.* Recall that $H = \mathbb{E}[g]\mathbb{E}[g]^\top$. The rotated Hessians are defined as:

$$H_U = (I \otimes U^\top)\mathbb{E}[g]\left((I \otimes U^\top)\mathbb{E}[g]\right)^\top, \tag{59}$$

$$H_{U,V} = (V^\top \otimes U^\top)\mathbb{E}[g]\left((V^\top \otimes U^\top)\mathbb{E}[g]\right)^\top. \tag{60}$$

For any rank-1 matrix $M = zz^\top$, $(1,1)$-norm satisfies $\|M\|_{(1,1)} = \|zz^\top\|_{(1,1)} = \|z\|_1^2$. By Lemma F.1, $(I \otimes U^\top)\mathbb{E}[g] = \text{vec}(\Sigma V^\top)$, $(V \otimes U^\top)\mathbb{E}[g] = \text{vec}(\Sigma)$ holds. Thus, proving the theorem is equivalent to showing $\|\Sigma\|_{(1,1)} \le \|\Sigma V^\top\|_{(1,1)} \le \|U\Sigma V^\top\|_{(1,1)}$.

Since $H$ is rank-1 and we assume it admits an exact Kronecker product structure, $\mathbb{E}[G]$ must also be rank-1. Let $\mathbb{E}[G] = \sigma_1 u_1 v_1^\top$, where $u_1$ and $v_1$ are the first columns of $U$ and $V$, respectively, and $\sigma_1$ is the singular value.

Since $U$ and $V$ are orthogonal, $\|u_1\|_2 = \|v_1\|_2 = 1$. By the norm inequality $\|x\|_1 \ge \|x\|_2$, we have $\|u_1\|_1 \ge 1$ and $\|v_1\|_1 \ge 1$. Therefore, the following inequalities hold:

$$
\begin{aligned}
\|\Sigma\|_{(1,1)} &= \sigma_1 \\
&\le \sigma_1 \|v_1\|_1 = \|\Sigma V^\top\|_{(1,1)} \\
&\le \sigma_1 \|v_1\|_1 \|u_1\|_1 = \|u_1 \sigma_1 v_1^\top\|_{(1,1)} = \|U\Sigma V^\top\|_{(1,1)}.
\end{aligned}
$$

This implies $\|H_{U,V}\|_{(1,1)} \le \|H_U\|_{(1,1)} \le \|H\|_{(1,1)}$. Furthermore, the diagonal form $\|H_{U,V}\|_{(1,1)}$ attains the global minimum. $\qquad\square$

## G. Connection to Recent Optimizers

In this section, we establish a connection between the design axes of `eigenbasis-estimation` and existing modern optimizers to contextualize our framework within the current literature. While our methodology shares conceptual roots with several recent optimizers, we introduce `eigenbasis-estimation` as a unified abstraction designed to isolate the effects of Hessian geometry from other implementation variables, such as momentum accumulation space or the use of power iteration instead of exact Singular Value Decomposition (SVD).

Our 2nd strategy is closely related to SOAP (Vyas et al., 2025) and EShampoo (Eschenhagen et al., 2025). However, a key technical distinction lies in the accumulation space of the optimizer states and the timing of the basis update. Unlike the official implementation of SOAP, which accumulates the first momentum in a rotated space and updates the eigenbasis after the parameter update step, our 2nd approach accumulates the first momentum in the original space and refreshes the basis before the optimization step. While accumulation in the rotated space requires projecting the momentum back and forth between bases during updates, accumulation in the original space does not require this additional computation. Furthermore, while EShampoo relies on SVD to compute eigenvectors, our implementation uses a single step of power iteration and QR decomposition to approximate them.

Similarly, our 1st strategy shares commonalities with GaLore (Zhao et al., 2024), LDAdam (Robert et al., 2025) (for unilateral), and the SVD-rotated AdamW variant discussed in Zhang et al. (2025a) (for bilateral). The primary departure from these methods is our use of the momentum matrix $M_t$ as the source for basis estimation, whereas GaLore and AdamW-SVD typically derive their rotation matrices from the instantaneous, noisy gradient $G_t$. Furthermore, unlike GaLore and AdamW-SVD—which accumulate momentum in a rotated space and rely on exact SVD—our approach maintains accumulation in the original space and utilizes power iteration. Finally, because `eigenbasis-estimation` remains strictly full-rank, it avoids the complex error-buffer mechanisms found in LDAdam.

By standardizing the optimizer's internal mechanics, we can systematically analyze how different basis estimation strategies interact with the loss landscape and gradient delay without being confounded by the implementation details of individual optimizers.

## H. Memory Overhead Analysis

In this section, we analyze the memory overhead of basis rotation in a realistic training setup. As described in Section 3.2, our `eigenbasis-estimation` framework spans two design axes, the approximation source $\mathcal{S} \in \{1\text{st}, 2\text{nd}\}$ and the rotation geometry $\mathcal{G} \in \{\text{Unilateral}, \text{Bilateral}\}$, yielding four concrete strategies. We quantify the memory cost of each strategy both in terms of theoretical complexity and concrete GB usage on a realistic LLM training setup.

For a single weight matrix $W \in \mathbb{R}^{m \times n}$, basis rotation introduces two sources of additional memory beyond the standard Adam states: (i) the rotation matrices used to diagonalize the Hessian approximation, and (ii) the second-moment statistics required when $\mathcal{S} = 2\text{nd}$. For the bilateral strategy ($\mathcal{G} = \text{Bi}$), both $U \in \mathbb{R}^{m \times m}$ and $V \in \mathbb{R}^{n \times n}$ are stored, incurring $m^2 + n^2$ parameters; for the unilateral strategy ($\mathcal{G} = \text{Uni}$), only the rotation along the smaller dimension is stored, reducing the cost to $\min(m,n)^2$. When $\mathcal{S} = 2\text{nd}$, the empirical Fisher factors $L = GG^\top \in \mathbb{R}^{m \times m}$ and $R = G^\top G \in \mathbb{R}^{n \times n}$ must be maintained as buffers, adding the same cost as the rotation matrices. In contrast, $\mathcal{S} = 1\text{st}$ reuses the existing momentum buffer $M_t$ to estimate the eigenbasis, introducing *no* additional moment storage.

Table 2 reports both the theoretical per-matrix overhead and concrete GB usage on Llama-3-8B, which has hidden dimension $h = 4096$ and MLP intermediate dimension $4h_{\text{int}} = 14336$. Attention projection matrices are square ($4096 \times 4096$), while MLP projection matrices are rectangular ($4096 \times 14336$). All buffers are stored in FP32 (4 bytes per element), consistent with standard mixed-precision training where optimizer states are kept in full precision.

*Table 2.* Memory overhead of basis rotation strategies on Llama-3-8B ($h = 4096$, $4h_{\text{int}} = 14336$). Memory is reported in GB per weight matrix, assuming FP32 storage. Attention matrices are $4096 \times 4096$; MLP matrices are $4096 \times 14336$.

| $\mathcal{S}$ | $\mathcal{G}$ | Rotation | Moments | Mem (Attn) | Mem (MLP) |
|---|---|---|---|---|---|
| 2nd | Bi | $m^2 + n^2$ | $m^2 + n^2$ | 0.25 | 1.66 |
| 2nd | Uni | $\min(m,n)^2$ | $\min(m,n)^2$ | 0.13 | 0.13 |
| 1st | Bi | $m^2 + n^2$ | – | 0.13 | 0.83 |
| 1st | Uni | $\min(m,n)^2$ | – | 0.06 | 0.06 |

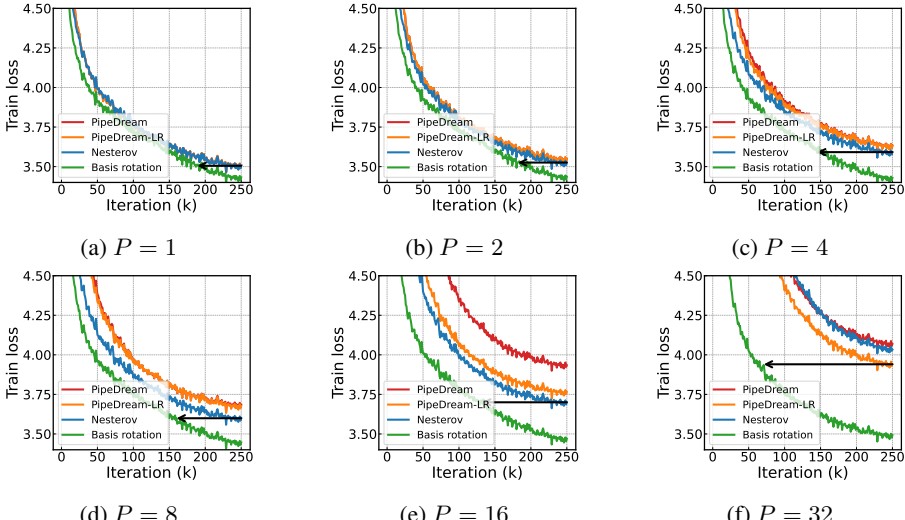

*Figure 12.* Comparison of each method for different number of stages $P$. The gap between basis rotation and baselines increases with larger $P$.

A few observations are in order. First, the bilateral variants ($\mathcal{G} = $ Bi) incur substantially more memory than their unilateral counterparts on rectangular MLP matrices, because they store both $U \in \mathbb{R}^{m \times m}$ and $V \in \mathbb{R}^{n \times n}$ rather than a single rotation along the smaller dimension. Second, switching from $\mathcal{S} = $ 2nd to $\mathcal{S} = $ 1st halves the overhead by eliminating the $L$ and $R$ buffers and reusing the existing momentum matrix $M_t$ for eigenbasis estimation. Third, the most memory-efficient strategy ($\mathcal{S} = $ 1st, $\mathcal{G} = $ Uni) requires only $\min(m, n)^2$ parameters per matrix—for an MLP layer with $m = 4n$, this is approximately $7.5\%$ relative to Adam's $4mn$ optimizer states ($m$- and $n$-shaped first and second moments stored in FP32)—while still delivering more than a $40\%$ speedup over the best-performing baseline (see Figure 8).

# I. More Experimental Results

**Full Results of Section 4.2** We plot the full results of Figure 5 in Figures 12 to 14. As seen in Figure 12, the gap between basis rotation and baselines becomes larger with increasing number of stages $P$. We also find that the slowdown, defined as the iteration ratio required to reach target loss for $P = 32$ relative to $P = 1$, becomes significantly smaller for basis rotation (see Figure 13). We note that this slowdown becomes smaller with more accurate `eigenbasis-estimation` strategies. Finally, we plot the full results of Figure 6 in Figure 14 where we increase the number of stages by increasing the number of Transformer blocks. While baseline methods invalidate the standard scaling law with increasing loss for larger models, basis rotation, especially with high-fidelity `eigenbasis-estimation` strategies, recovers the scaling law by gradually decreasing the loss for larger models.

**Full Results of Section 4.3** We plot the full results of Figure 8 in Figure 16. We note that the high-fidelity estimation strategy consistently outperforms its memory-efficient counterpart across all number of stages, with a widening gap for larger stages. We also note that even the least accurate estimation strategy (1st/unilateral) consistently outperforms the best-performing baseline with a widening gap for larger stages.

**Results with weight prediction** We also evaluate the performance of basis rotation when employing PipeMare-style weight prediction (Yang et al., 2021) as an alternative strategy for reducing memory overhead. The results, shown in Figure 15, demonstrate that basis rotation maintains its performance advantage and remains significantly more robust than baseline methods even when using approximated weight versions. This consistent trend further underscores the resilience of our approach to the gradient inaccuracies typical of asynchronous pipeline parallelism.

**Stage-aware basis rotation implementation and more results** Here, we provide implementation details and additional experimental results for the stage-aware basis rotation strategy. Under a fixed computational budget, the subspace update frequency at each pipeline stage is allocated proportionally to the corresponding gradient delay, such that the overall

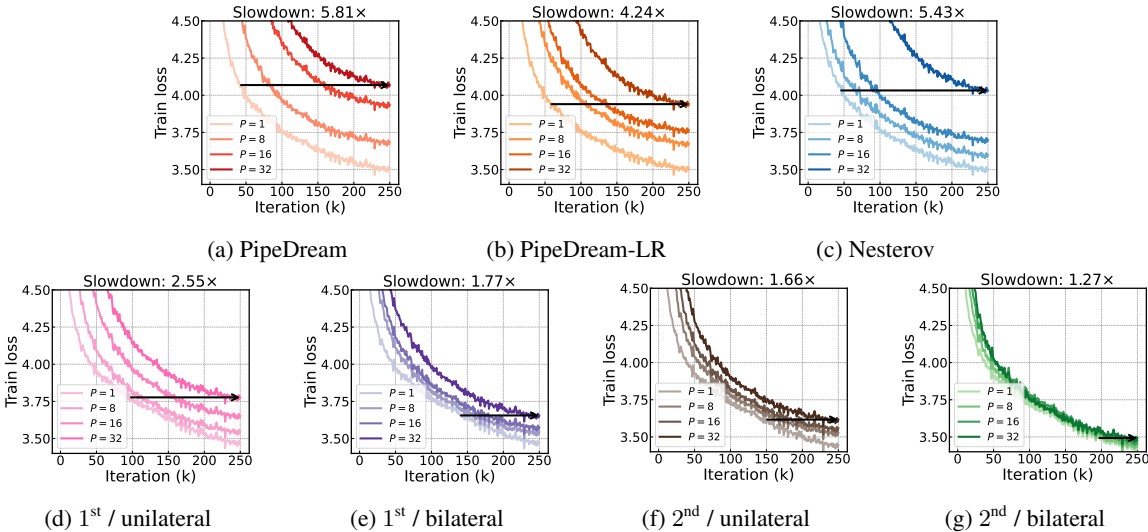

*Figure 13.* Slowdown for different methods when increasing the number of stages $P$ for baselines (a-c) and basis rotation (d-g). Here, slowdown is defined as the iteration ratio required to reach target loss for $P = 32$ relative to $P = 1$.

computational cost remains constant across configurations. Specifically, let $\tau$ denote the gradient delay at the stage. We define the update frequency $f$ at each stage as a function of $\tau$ as the following scheduling rule:

$$mid = \left\lfloor \frac{P}{2} \right\rfloor - 1$$

$$n = \begin{cases} mid - \tau, & \text{if } \tau > mid \\ mid + 1 - \tau, & \text{if } \tau \leq mid \end{cases}$$

$$f = \left\lfloor \frac{f_0}{1 - \frac{n}{mid}} \right\rfloor$$

To further validate the role of basis rotation, we additionally evaluate a reversed stage-aware allocation strategy, in which the update frequency is inversely assigned with respect to gradient delay. As shown in Figure 17, this reversed strategy leads to inferior convergence behavior, confirming the effect of basis rotation for mitigating staleness.

**Results for Validation Loss** We also plot the validation loss in Figure 18. Here, we measure validation loss for 200 examples every 1000 iteration. We observe that the trend is very similar to the results for train loss.

**Experiments for Other Baselines** We also compare our approach against Delay Compensation (DC) algorithm (Zheng et al., 2017). DC was originally proposed for asynchronous data parallelism and later adapted to pipeline parallelism for small-scale vision models (Jang et al., 2023). Specifically, it uses a first-order Taylor expansion to approximate the fresh gradient $\nabla f(w_t)$ from the delayed gradient $\nabla f(w_{t-\tau})$:

$$\nabla f(w_t) \approx \nabla f(w_{t-\tau}) + \lambda \nabla f(w_{t-\tau}) \odot \nabla f(w_{t-\tau})(w_t - w_{t-\tau}),$$

where $\lambda \in [0, 1]$ is a hyperparameter to control the scale the compensation term and the diagonal empirical Fisher $\nabla f(w_{t-\tau}) \odot \nabla f(w_{t-\tau})$ serves as a diagonal approximation of the Hessian. We test $\lambda \in \{0.04, 0.1, 0.5, 1.0\}$.

The results are plotted in Figure 19. We observe that DC does not effectively address delayed gradients for large delays and shows similar performance to PipeDream.

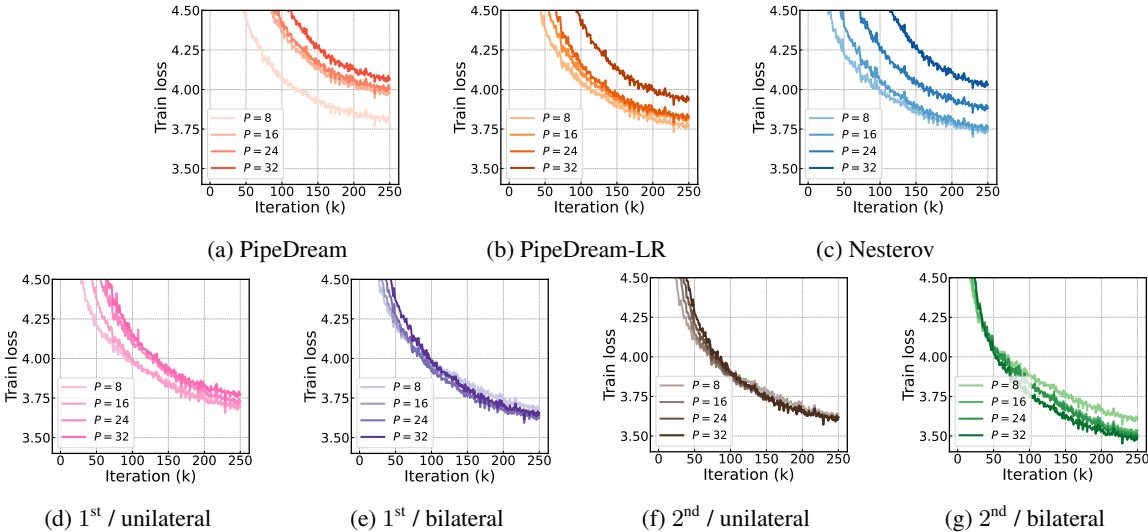

(a) PipeDream      (b) PipeDream-LR      (c) Nesterov

(d) $1^{st}$ / unilateral    (e) $1^{st}$ / bilateral    (f) $2^{nd}$ / unilateral    (g) $2^{nd}$ / bilateral

*Figure 14.* Performance of each method when increasing the number of stages by scaling the number of blocks for baselines (a-c) and basis rotation (d-g). While scaling the model rather increases the loss under asynchronous pipeline parallelism for baselines (a-c), scaling works well for basis rotation especially for $2^{nd}$/bilateral strategy (f).

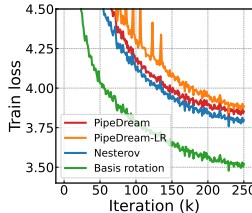

*Figure 15.* Performance of different methods when using PipeMare-style weight prediction (Yang et al., 2021) instead of weight stashing for $P = 32$. Basis rotation outperforms baselines with a large margin also in this setting.

*Table 3.* Slowdown of preconditioned methods at $P = 32$ relative to $P = 1$. Slowdown is defined as the ratio of iterations required to reach a fixed loss threshold at $P = 32$ versus $P = 1$. Methods with explicit basis alignment (SOAP and basis rotation) achieve substantially lower slowdown than those without (Muon and Scion), confirming that basis alignment is the primary factor for mitigating staleness under large delays.

| Method | Slowdown |
|---|---|
| PipeDream-LR | $5.43\times$ |
| Nesterov | $4.24\times$ |
| Muon | $1.53\times$ |
| Scion | $2.10\times$ |
| SOAP | $1.34\times$ |
| basis rotation | $1.27\times$ |

**Comparison with Preconditioned Methods**    We compare basis rotation against preconditioned optimizers that have recently demonstrated strong performance in standard (non-delayed) training settings: SOAP (Vyas et al., 2025), Muon (Jordan et al., 2024), and Scion (Pethick et al., 2025). While these methods were not originally designed for asynchronous pipeline parallelism, evaluating them in this setting provides insight into which algorithmic properties are most critical for delay robustness. We measure the slowdown of each method, defined as the ratio of iterations required to reach a fixed loss threshold at $P = 32$ relative to $P = 1$. The results are summarized in Table 3.

Muon and Scion, which do not perform explicit alignment with the Hessian eigenbasis, outperform standard baselines but still exhibit considerably higher slowdown than basis rotation. In contrast, SOAP achieves delay robustness close to basis rotation, as both methods share the principle of operating in a rotation aligned with the Hessian eigenbasis. These results support the central claim of our work: it is the basis alignment, rather than preconditioning per se, that is essential for

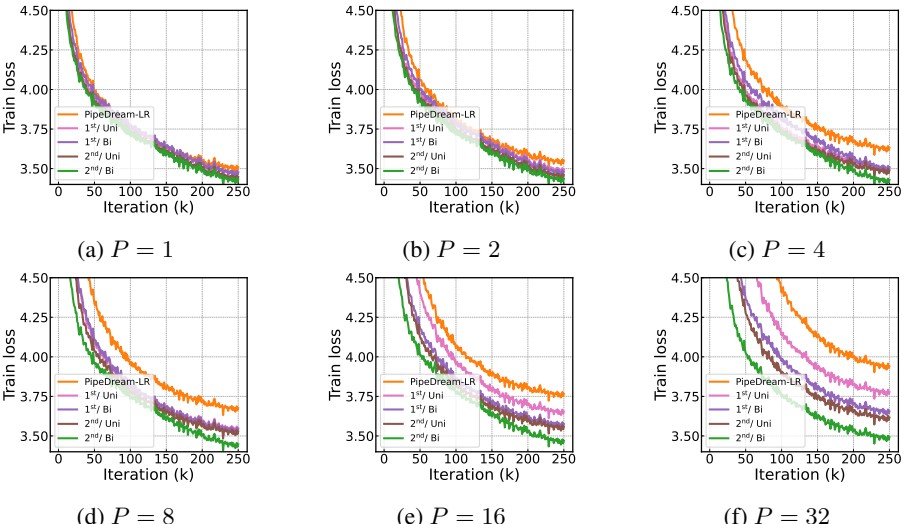

*Figure 16.* Comparison of different `eigenbasis-estimation` strategies for different number of stages $P$.

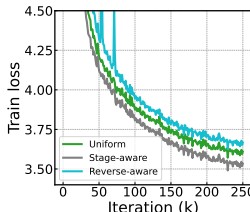

*Figure 17.* Stage-aware basis rotation and inverse stage-aware basis rotation. While stage-aware basis rotation outperforms uniform update frequency, inverse stage-aware basis rotation degrades the performance.

robustness to gradient delay in asynchronous pipeline parallelism.

We note that the minor implementation differences between basis rotation and SOAP—as described in Section 3.2 and Appendix G—were not intended to improve upon SOAP's performance. Rather, they were introduced to provide a controlled experimental environment for systematically analyzing different Hessian approximation strategies, free from optimizer-specific confounds.

**Results on 3B Model**    We evaluate basis rotation on an approximately 3B parameter model (3,047M parameters; 2688 embedding dimensions, 32 Transformer blocks) with $P = 32$ pipeline stages. We search over learning rates in $\{10^{-4}, 3 \times 10^{-5}\}$ with a 250K-step cosine schedule, and report partial training curves up to 82K iterations. As shown in Figure 20, basis rotation achieves the same training loss with $81.7\%$ fewer iterations than the best-performing baseline, further widening the advantage observed at 1B scale.

**Generalization to MoE**    We evaluate whether basis rotation generalizes beyond standard Transformer FFN layers to Mixture-of-Experts (MoE) architectures. Since MoE routing and expert computations are performed within a single GPU without altering the pipeline schedule, basis rotation can be applied to each expert's weight matrices independently without any modification to the pipeline scheduling logic. We train a 100M nanoMoE model [4] with eight experts and top-2 activation on the same language modeling setup described in Appendix D.2, and compare against the same baselines used in the main experiments.

The results in Figure 21 show that basis rotation achieves the lowest final training loss among all methods, reducing the iterations required to reach the same loss as the best-performing baseline by $46.8\%$. The consistent gains over baselines demonstrate that basis rotation effectively addresses gradient delay in MoE models, confirming that the benefits of our

---

[4] https://github.com/wolfecameron/nanoMoE

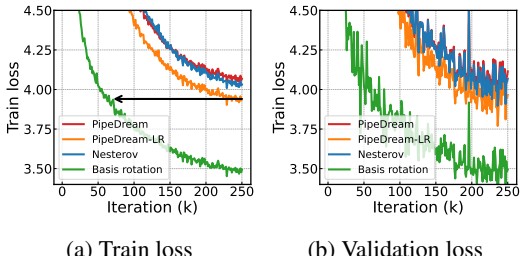

(a) Train loss          (b) Validation loss

*Figure 18.* Train loss (a) and validation loss (b) for different methods at $P = 32$. The trend in validation loss closely follows the trend in train loss.

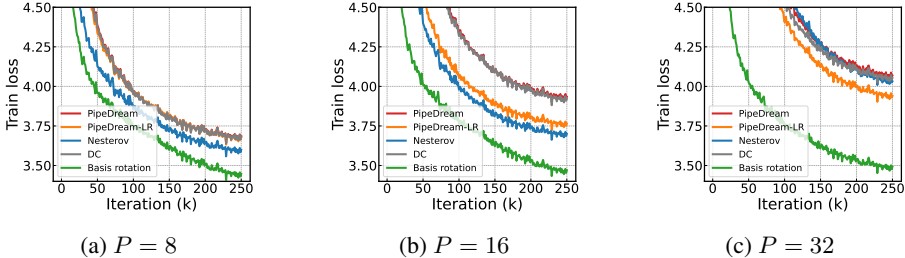

(a) $P = 8$                    (b) $P = 16$                    (c) $P = 32$

*Figure 19.* Comparison of each method for different number of stages $P$ including Delay Compensation (DC) algorithm (Zheng et al., 2017) as baselines. DC shows similar performance to PipeDream.

approach are not limited to dense Transformer architectures.

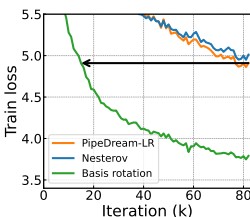

*Figure 20.* Performance of different methods at $P = 32$ for a $\approx$3B parameter model (2688 embedding dimensions, 32 Transformer blocks). Results are shown for partial training up to 82K iterations out of a 250K-step schedule. Basis rotation achieves the same training loss with $81.7\%$ fewer iterations than the best-performing baseline.

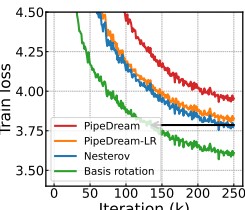

*Figure 21.* Performance of different methods on a 100M nanoMoE model (8 experts, top-2 activation). Basis rotation achieves the lowest final training loss and reduces the number of iterations required to reach the same loss as the best-performing baseline by $46.8\%$.

