# OpenReview forum: "Mitigating Staleness in Asynchronous Pipeline Parallelism via Basis Rotation"
_ICML.cc/2026/Conference — ICML 2026 regular_

### Official Review · Reviewer_EdTh · 2026-03-04

**Soundness:** 2
**Presentation:** 3
**Significance:** 2
**Originality:** 2
**Overall Recommendation:** 3
**Confidence:** 4

**Summary:**

The paper studies asynchronous pipeline parallelism. The authors first show through experiments that increasing the number of pipeline stages hurts convergence, since more stages mean larger delays in model updates and therefore more stale gradients.

Next, they point out that Adam is sensitive to rotations of the loss landscape. In particular, when the Hessian eigenvectors are not aligned with the standard coordinate axes, Adam performs worse. In this misaligned case, the negative effect of asynchrony becomes even stronger as the number of pipeline stages increases.

To address this, the authors propose a basis rotation framework. They estimate the Hessian and perform the optimization steps in a rotated coordinate system based on this estimate. They suggest several ways to approximate the Hessian. Their experiments show that the rotated approach improves convergence and performs better than other asynchronous pipeline parallel methods.

**Compliance With Llm Reviewing Policy:**

Affirmed.

**Final Justification:**

The rebuttal addressed my main concern, and therefore, I am increasing my score. Although the analysis is relatively simple and the main result is somewhat expected from my perspective, I think the paper does a good job of presenting and supporting it. Overall, I would be fine with the paper being accepted.

**Key Questions For Authors:**

See "Strengths And Weaknesses".

**Limitations:**

See "Strengths And Weaknesses".

**Strengths And Weaknesses:**

There already exist several preconditioned methods that are known to perform better than Adam, such as Shampoo [1], SOAP [2], and Muon [3]. It is therefore natural to expect that using a stronger preconditioned optimizer in an asynchronous pipeline parallel setting would lead to better performance than Adam.

The paper’s proposed method is very close to SOAP [2], with only minor modifications. The main strength of the paper is that it verifies, through experiments, that such a preconditioned approach improves performance in the asynchronous pipeline setting.

Let me now elaborate on the weaknesses.

1. **Weak theoretical contribution**.
    The theoretical analysis is limited. The main result studies convergence of Adam with delay, assuming a constant delay. In practice, different pipeline stages have different delays (see [4]), but this is not reflected in the analysis. Moreover, the analysis is conducted in a data-parallel regime and does not explicitly capture the structure of pipeline parallelism.

    The proof largely follows [5], with the only modification being the inclusion of a delay term, which results in an additional delay factor in the bound. This extension is straightforward.

    The theorem mainly serves to show that delay interacts with the basis misalignment coefficient, amplifying its effect. However, the paper does not provide a corresponding convergence analysis for the rotated method. In particular, it is unclear from theory how or why the proposed rotation improves the convergence rate.

2. **Limited novelty compared to existing methods**.
    The paper’s proposed method is very similar to SOAP [2], differing mainly in the update order. There is no clear motivation explaining why these modifications should lead to improved performance.

    Furthermore, SOAP itself is not included as a baseline in the experiments. It would be important to compare directly against SOAP, as well as other strong preconditioned methods such as Muon [3] or Scion [6]. It is plausible that simply applying one of these existing methods in the same setting could achieve similar or better results.

---

[1] Gupta, Vineet, Tomer Koren, and Yoram Singer. "Shampoo: Preconditioned stochastic tensor optimization." International Conference on Machine Learning. PMLR, 2018.

[2] Vyas, Nikhil, et al. "Soap: Improving and stabilizing shampoo using Adam." arXiv preprint arXiv:2409.11321 (2024).

[3] Jordan, K., Jin, Y., Boza, V., Jiacheng, Y., Cecista, F., Newhouse, L., and Bernstein, J. Muon: An optimizer for hidden layers in neural networks, 2024b. URL https: //kellerjordan.github.io/posts/muon/.

[4] Ajanthan, Thalaiyasingam, et al. "Nesterov method for asynchronous pipeline parallel optimization." arXiv preprint arXiv:2505.01099 (2025).

[5] Xie, Shuo, Mohamad Amin Mohamadi, and Zhiyuan Li. "Adam Exploits $\ell_\infty $-geometry of Loss Landscape via Coordinate-wise Adaptivity." arXiv preprint arXiv:2410.08198 (2024).

[6] Pethick, Thomas, et al. "Training deep learning models with norm-constrained lmos." arXiv preprint arXiv:2502.07529 (2025).

---

> ### Author Rebuttal · Authors · 2026-03-31
>
> ## [W1] Weak theoretical contribution
>
> **Constant delay is useful.**
>
> In async PP, delays are bounded by the maximum at stage 1; constant delay thus represents the worst-case. Even [1] (cited by the reviewer) uses this setup. Stage-dependent delay only tightens constant factors, and does not change the rate itself or the interaction between delay and smoothness as seen in [2].
>
> **Basis rotation (BR) improves convergence.**
>
> Adam with a fixed rotation is mathematically equivalent to Adam in the rotated space (L216-252); thus Thm 2.3 applies to the rotated method. Per Thm 2.3, a lower $C$ (locally associated with the Hessian (1,1)-norm) suppresses the adverse effects of delay (L190-212). Thm 3.1 proves BR reduces this norm, lowering $C$ and mitigating the impact of delay on convergence. Empirically, BR reduces the norm from 0.54 to 0.12, validating Thm 3.1.
>
> **Our results render reasonable contributions.**
>
> Thm 2.3, building on [3], is the first to characterize the interaction between delay and basis misalignment. Linked with Thm 3.1, it proves how BR enhances delay robustness. This logical sequence—from identifying the root cause of degradation to proving a solution—is believed to be a novel contribution unexplored in the literature.
>
> ---
>
> ## [W2] Limited novelty
>
> **It is beyond obvious expectations.**
>
> Translating the success in zero-delay settings to the async setting sounds intuitive, but is not always true due to complex delay interactions. For example, momentum is effective in standard settings but exacerbates gradient staleness in async [4]. Moreover, expecting and substantiating are strictly two different things. We identified why coordinate-wise adaptive optimizers are vulnerable to delay and demonstrated effective solutions, dramatically reducing iterations by 76.8% for a 1B model. Such an effort and realization are neither obviously expected nor previously conducted.
>
> **Experiments on preconditioned methods show basis alignment is the key factor.**
>
> Following the reviewer's suggestion, we evaluated SOAP, Muon, and Scion at P=1 and 32, measuring their slowdown.
> |Method|Slowdown|
> |-|-|
> |PipeDream-LR|5.43x|
> |Nesterov|4.24x|
> |Muon|1.53x|
> |Scion|2.10x|
> |SOAP|1.34x|
> |BR|1.27x|
>
> While Muon and Scion—which lack explicit basis alignment—outperform standard baselines, they show degradation compared to BR or SOAP. Conversely, SOAP achieves delay robustness similar to BR, as both share the rotation principle. These results support our core argument that basis alignment is essential for mitigating staleness.
>
> Here, we clarify that minor modifications made to BR relative to SOAP were not meant for performance gain. As explained in Sec. 3.2 and App. E, this was to provide a controlled environment to systematically analyze various Hessian approximations by excluding optimizer-specific variables. While we initially omitted preconditioned methods as baselines as they were unexplored in prior async PP literature, we will add these results to the revision.
>
> **We developed a stage-aware BR to tailor it for async PP and further enhance performance.**
>
> We tailored BR for async PP via a stage-aware update frequency which performs more precise basis alignment via more frequent updates in the earlier stages (with larger delays). Specifically, we allocate more updates to earlier stages: $f(i) \approx \frac{f_0}{2(1 - i/P)}$ ($f_0$: uniform freq., $P$: \# of stages, $i$: stage index). [Figure](https://anonymous.4open.science/r/icml1609/stageaware.pdf) shows that stage-aware BR achieves 29.2% speedup compared to standard BR. This directly validates our claim regarding the interaction between delay and basis misalignment.
>
> ---
> ## Closing remark
>
> We appreciate the reviewer’s constructive feedback, which helped us improve our work (e.g., stage-aware BR, Hessian norm reduction). The significance of our work are: **[S1]** identifying a previously unexplored “*real scalability bottleneck*” in async PP (`13qq`), **[S2]** providing a “*compelling diagnosis*” and remedy for basis misalignment (`UFmr`), and **[S3]** offering a “*potentially impactful*” path for large-scale async PP (`UFmr`). Furthermore, our originality lies in: **[O1]** establishing a new paradigm beyond "*treating staleness as something to predict or compensate*" [1, 5] by addressing its “*root cause*” (`13qq`), and **[O2]** revealing that the “*failure mode at large pipeline depth is not just generic gradient staleness, but the interaction*” between delay and basis misalignment (`UFmr`). We hope this resolves initial concerns and kindly ask for a positive reconsideration of the value of our work.
>
> ---
>
> [1] Nesterov Method for Asynchronous Pipeline Parallel Optimization\
> [2] Fully Decoupled Neural Network Learning Using Delayed Gradients\
> [3] Adam Exploits ℓ∞-geometry of Loss Landscape via Coordinate-wise Adaptivity\
> [4] Taming Momentum in a Distributed Asynchronous Environment\
> [5] PipeMare: Asynchronous Pipeline Parallel DNN Training

---

> > ### Author Rebuttal · Reviewer_EdTh · 2026-04-03
> >
> > Thank you for the detailed rebuttal and for adding further experiments. However, I am still not fully convinced, and my main concerns remain.
> >
> > 1. **On the usefulness of the constant-delay model.**
> >
> > I understand the argument that constant delay can be viewed as a useful worst-case abstraction. However, your own rebuttal seems to suggest that allowing stage-dependent treatment is in fact important in pipeline parallelism, since you later introduce a stage-aware BR specifically to exploit the difference between stages and report a significant improvement from doing so. This seems to support my original concern that modeling all delays as identical may miss an important part of the structure of the problem.
> >
> > More importantly, even if using a constant-delay model only affects constants, this still does not fully address my concern that the theory does not appear to match the actual method or setting of interest. The analysis reads much more like a standard delayed optimization analysis in a data-parallel setting, rather than an analysis that genuinely captures the pipeline-parallel structure. So my concern is not only that the delay model is simplified, but that the paper seems to analyze a different abstraction than the method actually studied.
> >
> > 2. **On whether Theorem 2.3 really applies to the rotated method.**
> >
> > I am also not convinced by the claim that Theorem 2.3 automatically applies to the rotated method. It is not even fully clear from the presentation which exact method Theorem 2.3 is analyzing. As written, it seems to analyze a standard Adam-type method, without explicitly incorporating any rotation. I also do not see the rotation appearing in the proof of the theorem itself.
> >
> > So the key step in your rebuttal, namely that Adam with a fixed rotation is mathematically equivalent to Adam in the rotated space and therefore Theorem 2.3 directly applies, needs to be shown much more rigorously. At present, this feels asserted rather than established. If this equivalence is central to the contribution, then it should be made explicit in the paper with a precise mathematical statement and proof, rather than being left implicit. Otherwise, it is difficult to assess whether the theorem truly explains the behavior of the rotated method or only that of the unrotated baseline.
> >
> > 3. **On the additional experiments with preconditioned methods.**
> >
> > Thank you for adding these experiments. As expected, methods such as Muon and SOAP also perform well in this setting, which is a useful context. However, I am still surprised by the gap between your method and SOAP, especially since the difference appears relatively small.

---

> > > ### Author Response · Authors · 2026-04-06
> > >
> > > ## On the Usefulness and Extension of the Constant-Delay Model
> > >
> > > **The success of stage-aware BR does not imply limitations of the constant delay model.**
> > >
> > > While the simplified model may not capture all nuances of stage-aware BR, it remains a “*useful worst-case abstraction*” and is a standard model in async PP [1-3]. Thus, Thm 2.3 holds significance for first identifying the root cause of degradation: the interaction between delay and basis misalignment. Crucially, stage-aware BR was directly motivated by the key insight from constant delay analysis: larger delays amplify the adverse effects of basis misalignment. This new method's success proves the practical power of our previous analysis.
> > >
> > > **We provide a proof sketch for stage-dependent delay to support the validity of stage-aware BR.**
> > >
> > > Nevertheless, we agree with the reviewer’s suggestion that reflecting the structure of async PP would enhance our rigor. Thus, we extend Thm 2.3 to stage-dependent delays, with a proof sketch below:
> > >
> > > Let $S_1,\dots,S_K$ be stages. For coordinate $i \in S_k$, the delay is $τ_i=K-k$ and the update is   $x_{t+1,i}=x_{t,i}-η\frac{g_{t_i,i}}{\sqrt{v_{t_i,i}}}$ (ϵ omitted) where $t_i=t-τ_i$.
> > >
> > > **STEP 1.** L840 handling
> > >
> > > In Eq. (1) of L840, $τ$ is replaced by $τ_i$. Bounds for (A),(C) follow similar steps with Lem. C.3,C.2. For (B), defining $A_t,B_t$ as in L860 by replacing $t-τ$ with $t_i$, Cauchy-Schwarz gives:$$η\sum_{t=1}^{T+K}A_tB_t\leη\sqrt{\sum_tA_t^2}\sqrt{\sum_{t,i}\frac{g_{t_i,i}^2}{v_{t_i,i}}}.$$
> > >
> > > Following L884–918 gives$$\sum_tA_t^2\le\sum_{t,i}H_i^2τ_i\sum_{k=0}^{τ_i-1} ||x_{t-τ_i-k+1}-x_{t-τ_i-k}||\_2^2\le\sum_iH_i^2 τ_i^2\sum_{t=1}^{T+K}||\frac{g_{t_i}}{\sqrt{v_{t_i}}}||\_2^2\le\left(\sum_i H_i^2τ_i^2\right)\sum_{t,j}\frac{g_{t,j}^2}{v_{t,j}}.$$These sequentially follow from Assumption 2.2 with the norm and Jensen's inequalities, the fact that each $\frac{g_{t_i}}{\sqrt{v_{t_i}}}$ is summed at most $τ_i$ times, and $||\frac{g}{\sqrt{v}}||\ge0$.
> > >
> > > **STEP 2.** Final result
> > >
> > > Now, $η∑_{t=1}^{T+K}A_tB_t$ is bounded as in (2), replacing $τ\sqrt{∑_iH_i^2}$ with $\sqrt{∑_iH_i^2 τ_i^2}$. Thus, we recover the form of Thm 2.3 with $τ':=\sqrt{\frac{∑\_iH_i^2τ_i^2}{∑\_iH_i^2}}=\sqrt{\frac{∑\_{k}(K-k)^2∑\_{i\in S_k}H_i^2}{∑_iH_i^2}}$, instead of $τ$.
> > >
> > > The result shows that stage-dependent delays tighten the constant factor ($τ'≤K$, max delay) while preserving the fundamental multiplicative effect between delay and misalignment. Also, $τ'$ validates that reducing misalignment in earlier stages ($∑_{i\in S_k}H_i^2$) via stage-aware BR effectively minimizes the bound. Importantly, this proof sketch highlights the extensibility of our theoretical framework, rather than making it obsolete. We thank the reviewer for the constructive feedback and will include the full proof in the revision.
> > >
> > > ---
> > >
> > >  ## On Thm 2.3 & Rotated Method
> > >
> > > **Adam with a fixed rotation is equivalent to Adam in the rotated space.**
> > >
> > > For a sequence $x=(x_i)\_{i=1}^t$, let $m_t(x)=(1-β_1)∑\_iβ_1^{t-i}x_i,v_t(x)=(1-β_2)∑\_iβ_2^{t-i}x_i^2,g_i=∇f(w_i)$. Adam with rotation $U$ is:$$w_{t+1}=w_t-ηU\frac{m_t(U^\top g)}{\sqrt{v_t(U^\top g)}}.$$Let $z=U^\top w,f’(z)=f(Uz)$. Multiplying $U^\top$($U^\top U=I$) yields $$z_{t+1}=z_t-η\frac{m_t(U^\top g)}{\sqrt{v_t(U^\top g)}}.$$Using chain rule $ g_i':=∇\_{z_i}f’(z_i)=U^\top g_i$ recovers Adam in rotated space:$$z_{t+1}=z_t-η\frac{m_t(g')}{\sqrt{v_t(g')}}.$$We will add this to the Appendix.
> > >
> > > **Thm 2.3 applies to the rotated method.**
> > >
> > > By the above equivalence, convergence analysis for rotated Adam with fixed $U$ is identical to applying Thm 2.3 to $f’$. Rotation $U$ does not appear in the proof as it only replaces $f$ with $f’$ and changes $C$. Thus, Thm 2.3 applies to Adam across any coordinate system with rotation only scaling $C$.
> > >
> > > **Our theory provides insight into the delay robustness of BR.**
> > >
> > > As $C$ locally corresponds to the Hessian (1,1)-norm, Thm 3.1 connects to our convergence analysis by proving that BR minimizes this norm. Together, Thm 2.3 and 3.1 provide a rigorous framework explaining the delay robustness of BR.
> > >
> > > ---
> > >
> > > ## On the Gap between BR and SOAP
> > >
> > > **The gap stems from the numerical instability of SOAP’s official implementation.**
> > >
> > > We hypothesize it is due to the accumulated FP round-off errors of SOAP’s per-step round-trip projection ($QQ^\top M$). Together with noise from delay, it even causes divergence at long update intervals. BR avoids redundant projections and ensures numerical stability. But again, this minor change was not meant for outperforming SOAP.
> > >
> > > ---
> > >
> > > ## Closing remark
> > >
> > > We appreciate the in-depth review. We hope our discussion and stage-aware BR—reinforcing our significance [S1-3] and originality [O1-2]—resolve the reviewer’s concerns and merit a reconsideration of the score.
> > >
> > > ---
> > >
> > > [1] Nesterov Method for Asynchronous Pipeline Parallel Optimization\
> > > [2] PipeMare: Asynchronous Pipeline Parallel DNN Training\
> > > [3] Pipelined Backpropagation at Scale: Training Large Models without Batches

---

### Official Review · Reviewer_UFmr · 2026-03-11

**Soundness:** 3
**Presentation:** 3
**Significance:** 3
**Originality:** 3
**Overall Recommendation:** 4
**Confidence:** 3

**Summary:**

This paper studies asynchronous pipeline parallelism for LLM training and argues that its main failure mode at large pipeline depth is not just generic gradient staleness, but the interaction between delayed gradients and Adam under basis misalignment between the Hessian eigenbasis and the coordinate basis. The proposed fix is basis rotation to estimate a rotated coordinate system so adaptive updates align better with curvature (estimate a rotated basis from gradient-derived curvature surrogates, run Adam-like preconditioning there, and rotate updates back). The authors provide a set of language-model pretraining experiments up to roughly 1B parameters, reporting large iteration savings versus PipeDream, PipeDream-LR, and Nesterov-style async baselines.

**Compliance With Llm Reviewing Policy:**

Affirmed.

**Final Justification:**

The authors have promised to incorporate extensions and clarifications into the final version of the manuscript to ensure greater clarity and completeness. I maintain my positive score.

**Key Questions For Authors:**

- In the main text and figure caption, the trajectories are presented as SGD vs Adam. But Appendix B.1 states that the authors used AdaSGD so both methods could share the same learning-rate machinery. Could the authors clarify this?

- Could you clarify more about the "number of stages"? The paper reports experiments with up to 32 stages on the 95M model while Appendix B.2 says those experiments used at most 8 RTX 3090s. Similarly, the 1B model uses 24 stages with at most 6 A100s. Would multiple virtual stages must be sharing a physical GPU? If so, then stage depth, delay, utilization, and GPU-hours comparisons becomes more subtle and it would be helpful to explicitly explain this.

**Limitations:**

Yes.

**Strengths And Weaknesses:**

Soundness: Good. I do not see any obviously fatal contradictions in the high-level intuition or in the idealized version of Theorem 3.1.

Presentation: The mathematics appears (generally speaking) clear.

Significance/Originality: This is an interesting and potentially impactful paper. The diagnosis is compelling and the idea seems useful.


Some strengths/weaknesses:

- The paper makes a convincing case that delay grows with pipeline depth and can become a real scalability bottleneck for asynchronous pipelining, rather than a minor nuisance. I agree that this is a real problem. Stale gradients in asynchronous pipeline parallelism are important, and the empirical slowdown with pipeline depth is meaningful.

- The basis-misalignment explanation is intuitive and easy to understand.

- The main convergence theorem analyzes asynchronous Adam with $\beta_1 = 0$, while the main experiments use momentum settings closer to practice ($\beta_1 = 0.9$ for most methods and $0.99$ for Nesterov). So the theory does not really cover the deployed method, but it does support the paper’s intuition (even if not the actual implementation).

---

> ### Author Rebuttal · Authors · 2026-03-31
>
> We appreciate the reviewer’s positive comments and the recognition of our work as interesting, impactful, compelling, and useful. Our responses to the specific questions are provided below.
>
> ---
>
> ## [W1] $\beta_1=0$ in convergence analysis
> **We focused on $\beta_1 = 0$ to better illustrate the intuition while avoiding complexity**
>
> We thank the reviewer for recognizing our $\beta_1=0$ analysis supports the paper's intuition. Following [1], extending to $\beta_1>0$ is achievable via [2], which introduces $\frac{1}{1-\beta_1}$ polynomials into the bound. As this extension offers no fundamentally new insights, we focused on $\beta_1=0$ to prevent unnecessary complexity from obscuring our core arguments.
>
> **We provide a proof sketch for the extension**
>
> For completeness, we sketch the $\beta_1>0$ extension (due to the space limit, we omitted $\epsilon$ and denote $\bar{G}$ as $G$, $\tilde{v}$ as $v'$). In Eq. (1) of App. C, replacing gradient $g$ with momentum $m$ modifies the proof:
>
> **STEP 1.** Bounding $E\equiv\frac{1}{T}\mathbb{E}\sum_{i,t}\sum_{k=0}^{t-\tau-1}\beta_1^k\frac{G_{t-\tau-k,i}^2}{\sqrt{v'_{t-\tau,k+1,i}}}$ (instead of L955)
>
> Instead of Lemma C.3, we bound term (A) of Eq. (1) using the lower bound of $\mathbb{E}\frac{G_{t,i}m_{t,i}}{\sqrt{v_{t,i}}}$. This process is achieved via the proof technique of Lemma A.1 in [2]. It yields:
>
> $$\mathbb{E}\sum_t\frac{G_{t,i}m_{t,i}}{\sqrt{v_{t,i}}}+\mathcal{O}((\eta H_i+\sigma_i)\sum_t\frac{g_{t,i}^2}{v_{t,i}})\ge\frac{1-\beta_1}{2}\mathbb{E}\sum_{t,k}\beta_1^k\frac{G_{t-k,i}^2}{\sqrt{v'_{t,k+1,i}}}.$$
>
> Since $\sum_t\frac{m_{t}^2}{v_{t}}$ from terms (B) and (C) of Eq. (1) is also bounded by $\mathcal{O}(\sum_t\frac{g_{t}^2}{v_{t}})$, L955 becomes:
>
> $$E\le\mathcal{O}\Big(\frac{\Delta_0}{\eta T}+\frac{1}{T}\sum_i(\eta H_i+\sigma_i)\max_i\sum_t\frac{g_{t,i}^2}{v_{t,i}}\Big).$$
> Bounding the max term via Lemma C.2 confirms the rates for $T, \eta, \sigma, H$ identically match term E (L993).
>
> **STEP 2**. Bounding $\min_{\frac{T}{2}< t\leq T} \mathbb{E}\|\nabla f(x_t)\|_1$
>
> By Cauchy-Schwarz, $\min_{t}\mathbb{E}\|G_t\|_1$ is bounded by
>
> $$\frac{2}{T}\mathbb{E}\sum_{t,i,k}\beta_1^k|G_{t-k,i}|\le\sqrt{2E}\left(\frac{2}{T}\mathbb{E}\sum_{t,i,k}\beta_1^k\sqrt{v'_{t,k+1,i}} \right)^{\frac{1}{2}}\equiv 2\sqrt{EE'}.$$
>
> Lastly, from the recursive technique of Lemma 3.16 in [1], we can obtain $E'\le \mathcal{O}\Big(E+\mathbb{E}\sum_{i,t} \sqrt{v'_{t,i}}/T\Big)$. Applying Lemma C.4 bounds this by $\mathcal{O}(E + \sum_i \sigma_i + \frac{1}{T})$, matching L1016. Thus, the interaction structure between $\tau$ and $C$ remains identical.
>
> ---
>
> ## [Q1] AdaSGD instead of SGD
> **We used AdaSGD for fair comparisons.**
>
> Since Adam scales gradients per parameter, its effective learning rate (LR) differs from vanilla SGD. Thus, we used AdaSGD to control for LR and solely examine the effect of coordinate-wise adaptivity on delay.
>
> **Vanilla SGD makes it difficult to isolate the effect of coordinate-wise adaptivity.**
>
> As shown in [figure](https://anonymous.4open.science/r/icml1609/sgd_vs_adasgd.pdf), SGD's behavior varies drastically with LR. Consequently, comparing the trajectory and delay sensitivity of Adam against vanilla SGD would require a subjective and arbitrary selection of the SGD LR.
>
> **We will clarify this in the revised manuscript.**
>
> Labeling it as "SGD" was a notational convenience for intuitive reading. However, as stated in App. B.1, AdaSGD was indeed used. We will revise to use "AdaSGD" in the final version to prevent any confusion.
>
> ---
>
> ## [Q2] Clarification on “number of stages”
>
> **We used multiple virtual stages.**
>
> The reviewer is correct; we configured multiple virtual stages to share a single GPU due to limited hardware. For instance, a 32-stage experiment on eight 3090 GPUs placed four virtual stages per GPU.
>
> **Our comparative results remain valid.**
>
> First, stage depth and delay are identically preserved. Because pipeline scheduling and communication order remain exactly the same, the gap from gradient computation to parameter update is unchanged.
> Second, while absolute utilization and GPU hours may fluctuate, the comparative results remain strictly valid. Simulating multiple stages on a single GPU can increase processing time due to context-switching overhead. However, since these conditions applied equally to the baseline and our method, relative comparisons (e.g., the loss vs. GPU hour in Fig. 10a) remain completely valid.
> Ultimately, basis rotation's core improvement stems from drastically reducing the total iterations required to reach a target loss, not from decreasing per-iteration time. Thus, our conclusion—that the method overwhelmingly saves total GPU hours—holds true regardless of setup-specific overheads. We appreciate the reviewer’s insightful observation and will clarify these points in the final version.
>
> ---
> [1] Adam Exploits ℓ∞-geometry of Loss Landscape via Coordinate-wise Adaptivity\
> [2] A Simple Convergence Proof of Adam and Adagrad

---

> > ### Author Rebuttal · Reviewer_UFmr · 2026-03-31
> >
> > I thank the authors for their reply. Yes, it would be good to have the promised extension in the final version of the paper. I also appreciate their clarifications on Q1 and Q2. If these points will be clarified in the revision, I have no further comments. I will keep my positive score.

---

> > > ### Author Response · Authors · 2026-04-06
> > >
> > > We sincerely thank the reviewer for the acknowledgment and for the time spent reviewing our rebuttal. We are glad to hear that our responses and clarifications regarding the theoretical extension for $\beta_1 > 0$, the use of AdaSGD, and the experimental setup with virtual stages fully addressed your concerns. As promised, we will incorporate these extensions and clarifications into the final version of the manuscript to ensure greater clarity and completeness. We appreciate your constructive feedback, which has significantly improved the quality of our work.

---

### Official Review · Reviewer_13qq · 2026-03-11

**Soundness:** 3
**Presentation:** 3
**Significance:** 3
**Originality:** 3
**Overall Recommendation:** 4
**Confidence:** 3

**Summary:**

This paper focus on the problem of asynchronous pipeline parallelism for training large language models. It argues that the main reason of the performance degrades with delay is not the delay itself, but that Adam’s per‑parameter scaling becomes ineffective when the Hessian eigenbasis is not aligned with the coordinate axes. To fix this, the authors propose “basis rotation”, which estimates the eigenbasis (using efficient approximations like Kronecker factorization) and rotate gradients and momentum into that basis before applying Adam, then rotate back. Experiments on models up to 1B parameters show that this method outperforms existing asynchronous baselines and even restores scaling laws.

**Compliance With Llm Reviewing Policy:**

Affirmed.

**Final Justification:**

I have read the response and will maintain my decision.

**Key Questions For Authors:**

1. While the paper mentions memory overhead, a rough estimate of the extra cost in a realistic low‑precision training setup would help practitioners judge the trade‑off.
2. The paper focuses on standard Transformer FFN layers. How does it generalize to MoE layers?

**Limitations:**

yes

**Strengths And Weaknesses:**

**Strengths**
1. The core idea is intuitive and well motivated: Instead of treating staleness as something to predict or compensate, this paper identifies the root cause as a geometric misalignment in the optimizer, which is vividly illustrated in the empirical observation.
2. The experimental evaluation is thorough: The proposed approach significantly improves the convergence speed over baselines methods, (both in terms of GPU time and number of iterations), across various model sizes, number of pipeline stages, and still generalizes without weight stashing to show robustness.
3. The benefit of the proposed method scales positively: Larger models actually benefit more, which is a promising sign for practical use in big LLM training.
3. The paper is clearly written.

**Weaknesses**
1. The max number of parameters of the validated models is limited to 1B.
2. The method is specifically designed for asynchronous pipeline parallelism with Adam. It is not clear how well it would transfer to other asynchronous settings or other optimizers.

---

> ### Author Rebuttal · Authors · 2026-03-31
>
> We thank the reviewer for insightful comments and constructive feedback. Please find our responses below.
>
> ---
>
> ## [W1] Validation for >1B models
>
> **We were unable to run experiments on models over 1B due to limited resources; we will provide 3B results in the revised version.**
>
> In an ideal environment where the number of GPUs at least matches the number of stages, training a 24-stage 1B model on 1B tokens requires ~54 GPU hours (at 10% MFU). However, our available resources were limited to six A100 GPUs, and we had to implement various techniques such as context switching, CPU offloading, and recomputation, to handle the full workload. For reference, the experiment in Fig 7b takes ~40 GPU days at 1B scale, and a 3B model would exceed 120 days. To our knowledge, the largest model previously explored in Async PP is around 1B [1], and our experiments were conducted with 2.5 times the token budget. Please note that we are in the process of securing institutional resources to include nanoGPT 3B results in the revised version.
>
> ---
>
> ## [W2] Extension to other settings
>
> **Basis rotation extends to other async settings.**
>
> Basis rotation mitigates delayed gradient regardless of specific delay structures, theoretically extending to async DP or RL. Nevertheless, we consider experiments on alternative asynchronous settings to be quite out of scope for the current study; we leave this for future investigation.
>
> **Basis rotation extends to other optimizers.**
>
> Basis rotation is applicable to any coordinate-wise adaptive optimizer. We validated its efficacy for Adafactor [2]. Applying basis rotation to Adafactor decreased the loss from 6.87 to 3.54. This dramatic improvement demonstrates that our framework is a general solution for coordinate-wise adaptive schemes sensitive to gradient staleness.
>
> **Exploiting the structure of async PP makes further enhancements.**
>
> Since the adverse effects of basis misalignment are amplified as the delay increases, we develop stage-aware basis rotation, which performs more precise basis alignment via more frequent updates in the earlier stages where the delay is larger. To achieve this while keeping the total computation budget constant, we allocate a higher update frequency to stages with larger delays. The stage-aware frequency $f(i)$ is calculated as $f(i) \approx \frac{f_0}{2(1 - i/P)}$ ($f_0$: uniform frequency, $P$: \# of stages, $i$: stage index). Experimental results at $P=32, f_0=100$ show that stage-aware basis rotation achieved an additional 29.2% speedup compared to standard basis rotation (see [figure](https://anonymous.4open.science/r/icml1609/stageaware.pdf)).
>
> ---
> ## [Q1] Memory overhead in a realistic setup
>
> **We calculated the memory overhead in a realistic setup.**
>
> For a weight $W \in \mathbb{R}^{m \times n}$, we calculated the theoretical and actual memory usages (GB) of different eigenbasis estimation strategies for Llama-3-8B. The results are shown below.
>
> | Strategy | Rotation | Moments | Total Complexity | Mem (Attn) | Mem (MLP) |
> | :--- | :--- | :--- | :--- | :--- | :--- |
> | 2nd / Bi | $m^2 + n^2$ | $m^2 + n^2$ | $2m^2 + 2n^2$ | 0.25 | 1.66 |
> | 2nd / Uni | $\min(m,n)^2$ | $\min(m,n)^2$ | $2\min(m,n)^2$ | 0.13 | 0.13 |
> | 1st / Bi | $m^2 + n^2$ | $-$ | $m^2 + n^2$ | 0.13 | 0.83 |
> | 1st / Uni | $\min(m,n)^2$ | $-$ | $\min(m,n)^2$ | 0.06 | 0.06 |
>
> Bi requires $m^2 + n^2$ for rotation because both $U \in \mathbb{R}^{m \times m}$ and $V \in \mathbb{R}^{n \times n}$ are stored. In contrast, Uni only stores either $U$ or $V$, reducing the overhead to $\min(m,n)^2$. 2nd requires storing second-moment statistics ($R=G^\top G \in \mathbb{R}^{n \times n}$, $L=GG^\top \in \mathbb{R}^{m \times m}$) whereas 1st does not require this additional memory.
>
> **We provide the following guidance for practitioners.**
>
> - Use 2nd/Bi when performance is the top priority.
> - Use Uni to reduce memory for MLP.
> - Use 1st/Uni in memory-constrained environments. It incurs only ~7.5% relative overhead compared to Adam’s $4mn$ (for MLP with $m=4n$) while still achieving a >40% speedup.
>
> ---
> ## [Q2] Generalization to MoE
>
> **Basis rotation generalizes well to MoE.**
>
> Basis rotation extends to MoE layers as MoE routing and expert computations are done within the same GPU without altering pipeline scheduling. We conducted experiments with 100M nanoMoE (8 experts, top-2 activation) and the results are shown below.
>
> | Method | Final loss |
> | :--- | :--- |
> | PipeDream-LR | 3.83 |
> | Nesterov | 3.79 |
> | Basis Rotation | 3.61 |
>
> Basis rotation significantly reduces the training loss in MoE compared to other baselines. Specifically, it reduces the number of iterations required to achieve the same training loss by 46.8% compared to the best-performing baseline (see [figure](https://anonymous.4open.science/r/icml1609/moe.pdf)).
>
> ---
>
> ## References
> [1] Nesterov Method for Asynchronous Pipeline Parallel Optimization \
> [2] Adafactor: Adaptive Learning Rates with Sublinear Memory Cost

---

> > ### Author Rebuttal · Reviewer_13qq · 2026-04-03
> >
> > I thank the authors for their response.  My concerns have been largely addressed. I am going to keep the positive score.
> > I would also appreciate it if the details regarding the memory overhead under different settings could be incorporated into the revised paper.

---

> > > ### Author Response · Authors · 2026-04-06
> > >
> > > We sincerely thank the reviewer for the positive feedback and for recognizing concerns as fully resolved. We are particularly grateful for your suggestion regarding the memory overhead analysis and the extension to MoE. As you requested, we will ensure that the detailed breakdown of memory overhead under various settings—including the trade-offs between different eigenbasis estimation strategies—is fully incorporated into the revised manuscript. We believe this will provide valuable guidance for practitioners. Thank you again for your insightful comments that helped improve the practical depth of our work.

---

### Decision · Program_Chairs · 2026-04-30

**Decision:**

Accept (regular)

**Comment:**

The paper identifies basis misalignment (not just staleness) as the root cause of Adam's degradation in asynchronous pipeline parallelism. The reviewers agree that the intuition is clear and well-supported. The empirical results are solid, with significant improvements over asynchronous baselines, with gains scaling positively to larger models. The paper is also well-structured, with a coherent story from toy examples to theory to large-scale experiments.

However, the reviewers found the method to be very close to SOAP, which was not included as a baseline. The paper does not convincingly justify why these modifications constitute a novel contribution. The analysis assumes constant delay and no momentum (β₁=0), while experiments use stage-dependent delays and momentum. No convergence analysis is provided for the rotated method itself. Some reviewers would also have liked validation on even larger models, but I recognize that this requires significant compute resources that are not available to all research teams.

I also find the idea that basis misalignment, and not staleness per se, is the problem is interesting and insightful.  The empirical results are solid, and the positive scaling with model size is promising. However, I share the reviewer concerns about limited novelty relative to SOAP and the gap between the theoretical analysis and the actual method. On the practical side, I would also have appreciated direct measurements during LLM training that show how the misalignment proxy increases with pipeline depth and how basis rotation reduces it in tandem with improved convergence, and it would have been nice with a more thorough discussion about failure modes that can occur when the eigenbasis estimation is inaccurate (e.g., during early training when statistics are noisy, or for layers with different curvature structure) and the impact this has on training.

Nevertheless, I suggest Weak Accept. The paper identifies a real, under-explored problem, provides a clear explanation, and demonstrates consistent empirical gains.